# GPan-LoRA: Gaussian Process Amortized Networks for Bayesian Low-Rank Adaptation in Large Language Models

**Weifeng Zhang** [* 1]  **Wenyuan Zhao** [* 1]  **Amir Hossein Rahmati** [1]  **Yucheng Wang** [1]
**Zhiyuan Wang** [1]  **Chao Tian** [1]  **Xiaoning Qian** [1 2 3]

## Abstract

Principled uncertainty quantification (UQ) is increasingly recognized as essential for trustworthy artificial general intelligence (AGI). Bayesian Low-Rank Adaptation (LoRA) provides a principled mechanism for uncertainty-aware fine-tuning of large language models (LLMs). However, existing techniques either face scalability constraints, e.g. Laplace-LoRA, or rely on approximate inference schemes that lead to poorly calibrated posterior uncertainty, often manifesting as overconfident predictions under distribution shift. To address this challenge, we propose GPan-LoRA, the first scalable Gaussian Process (GP)-based framework for Bayesian LoRA, which integrates neural network-based sparse GP approximations with amortized variational inference. By preserving the Bayesian function prior and posterior semantics intrinsic to GPs, GPan-LoRA achieves a faithful balance between computational scalability and principled UQ. Empirically, GPan-LoRA produces well-calibrated uncertainty that remains reliable under distribution shift, mitigating overconfident failures while preserving competitive task performance.

## 1. Introduction

Large Language Models (LLMs), such as GPT, Gemini, and their successors, have demonstrated remarkable capabilities in understanding and generating human-like text across vari-

---
[*]Equal contribution  [1]Department of Electrical & Computer Engineering, Texas A&M University, College Station, Texas, USA [2]Department of Computer Science & Engineering, Texas A&M University, College Station, Texas, USA [3]Applied Mathematics, Computing & Data Sciences, Brookhaven National Laboratory, Upton, New York, USA. Correspondence to: Weifeng Zhang <weifengzhang@tamu.edu>, Wenyuan Zhao <wyzhao@tamu.edu>.

*Proceedings of the 43rd International Conference on Machine Learning*, Seoul, South Korea. PMLR 306, 2026. Copyright 2026 by the author(s).

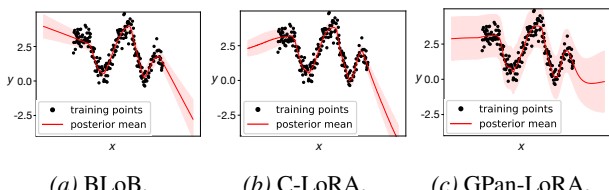

*Figure 1.* Fine-tuning a regression dataset with Bayesian LoRA. BLoB and C-LoRA exhibit low empirical coverage (red shadow) of confidence intervals, indicating overconfidence, while GPan-LoRA produces well-calibrated uncertainty estimates.

ous tasks (Wei et al., 2022; Teubner et al., 2023; Zhao et al., 2023; Chang et al., 2024; Majumder et al., 2024). However, the inherent uncertainty (Abdar et al., 2021) in their outputs poses significant challenges to reliable deployment in critical applications, such as healthcare, legal advisory, finance, and autonomous systems (Kasneci et al., 2023; Kaddour et al., 2023; Thirunavukarasu et al., 2023).

Bayesian learning naturally provides uncertainty quantification (UQ) by modeling predictive $P(\boldsymbol{y}|\boldsymbol{x}, \mathcal{D})$, which is the marginalization of posterior $P(\theta|\mathcal{D})$:

$$P(\boldsymbol{y}|\boldsymbol{x}, \mathcal{D}) = \int P(\boldsymbol{y}|\boldsymbol{x}, \theta)P(\theta|\mathcal{D})\mathrm{d}\theta. \qquad (1)$$

Exact posterior inference in LLMs is computationally infeasible due to their massive scale, making it challenging to accurately estimate uncertainty.

Low-rank adaptation (LoRA, Hu et al., 2022) is one of the most widely used parameter-efficient fine-tuning (PEFT) for LLMs. Recent research explores Bayesian inference under the LoRA mechanism to calibrate LLMs (Wang et al., 2023; Yang et al., 2023; Onal et al., 2024). For example, Bayesian Low-rank adaptation by Backpropagation (BLoB, Wang et al., 2024) jointly estimates the variational posterior of Bayesian neural networks (BNNs) throughout the entire fine-tuning stage. C-LoRA (Rahmati et al., 2025) further improves scalability with amortized variational low-rank parameters, allowing sample-dependent uncertainty.

Existing Bayesian LoRA models face a fundamental trade-off between scalability and UQ quality: Approaches that provide well-calibrated and expressive uncertainty estimates

often incur prohibitive computational costs (Balabanov & Linander, 2024), while scalable models typically rely on simplified or heuristic uncertainty representations that compromise UQ fidelity (Gal & Ghahramani, 2016; Boluki et al., 2020; Fan et al., 2021). As illustrated in Figure 1, when evaluated on a toy regression task, BLoB exhibits the issue of overconfidence during fine-tuning. Even with sample-dependent amortization, C-LoRA still struggles with overconfidence.

Gaussian processes (GPs), as a principled probabilistic approach, have been successfully combined with neural network (NN) heads in deep kernel learning (DKL, Wilson et al., 2016a;b). Similar overfitting trends were identified by Ober et al. (2021) in DKL, noting that while fully Bayesian models can help alleviate overfitting, they are not feasible for LLMs due to their extremely high computational and memory requirements. More recently, Matias et al. (2024) proposed amortized variational DKL (AVDKL) to mitigate the overfitting problem by using amortized inducing points and a parameter-sharing scheme.

Despite the strong UQ guarantees, integrating GP-based Bayesian modeling into PEFT frameworks such as LoRA for LLMs is highly non-trivial (Bradshaw et al., 2017). First, classical GPs scale poorly with both the number of data points $N$ and the input dimensionality $D$ (Williams & Rasmussen, 2006), while LLMs operate in regimes where both are extremely large. Although sparse and approximate GP methods alleviate the cubic dependence on $N$ and LoRA reduces the dimensionality to low rank $r$, applying sparse GPs directly to LoRA layers effectively results in very deep GPs (DGPs, Damianou & Lawrence, 2013), significantly increasing the complexity further (Salimbeni & Deisenroth, 2017). Second, classical GP inference is based on kernel matrix operations, including matrix inversion or linear solves (Titsias, 2009), which are fundamentally misaligned with the training and inference characteristics of LLMs. As a result, naively combining GPs with LoRA across all layers is prohibitively expensive and incompatible with the efficiency requirements of modern LLM fine-tuning.

To address the above challenge, we propose a new Bayesian framework of **Lo**w-**R**ank **A**daptation by **G**aussian **P**rocesses **a**mortized **n**etworks (**GPan-LoRA**). A variational sparse approximation, together with amortized inference, is applied to low-rank GP, offering a favorable scalability–UQ capability trade-off compared to prior GP-based Bayesian formulations, while maintaining the strong UQ properties intrinsic to GPs. We summarize the following contributions:

1. We propose GPan-LoRA, a scalable GP–based framework for Bayesian LoRA that enables principled UQ in LLMs with low computational overhead.

2. GPan-LoRA provides an end-to-end Bayesian framework that estimates uncertainty jointly with model

*Table 1.* Qualitative comparison of UQ methods in LoRA variants, where scalability reflects parameter efficient model and UQ quality captures the calibrated predictive uncertainty (✓: strong, △: moderate, ✗: weak).

| Method | UQ Quality | Scalability |
|---|---|---|
| MLE (Hu et al., 2022) | ✗ | ✓ |
| MCD (Gal & Ghahramani, 2016) | ✗ | ✓ |
| DeepEns (Wang et al., 2023) | △ | ✗ |
| LAP (Yang et al., 2023) | ✓ | ✗ |
| BLoB (Wang et al., 2024) | △ | △ |
| C-LoRA (Rahmati et al., 2025) | △ | ✓ |
| **GPan-LoRA** (ours) | ✓ | △ |

adaptation, mitigating the issue of overconfidence.

3. We demonstrate better UQ quality and scalability in our proposed GPan-LoRA across various LLM backbones and different use cases.

For reproducibility, our implementation is publicly available on Github[1].

## 2. Preliminaries

### 2.1. Bayesian LoRA

LoRA assumes a low-rank decomposition for the weight updates to fine-tune a pre-trained network (Hu et al., 2022). Specifically, instead of directly updating the full weight matrix $\boldsymbol{W} \in \mathbb{R}^{m \times n}$ in a linear layer, LoRA approximates the surrogate weight $\Delta \boldsymbol{W}$ as the product of two low-rank matrices $\boldsymbol{B} \in \mathbb{R}^{m \times r}$ and $\boldsymbol{A} \in \mathbb{R}^{r \times n}$:

$$\boldsymbol{h} = (\boldsymbol{W}_0 + \Delta \boldsymbol{W})\boldsymbol{x} = \boldsymbol{W}_0 \boldsymbol{x} + \boldsymbol{B}\boldsymbol{A}\boldsymbol{x}, \quad (2)$$

where $r \ll \min\{m, n\}$. The original pre-trained weights $\boldsymbol{W}_0$ are kept frozen during fine-tuning, and only the low-rank matrices $\boldsymbol{A}$ and $\boldsymbol{B}$ are optimized.

Bayesian LoRA variants introduce probabilistic priors over LoRA weights and perform variational inference (VI) on posterior, enabling uncertainty-aware PEFT directly in the adaptation layers (Yang et al., 2023). BLoB (Wang et al., 2024), as one of Bayesianized LoRA frameworks, simultaneously updates the mean and covariance of $\Delta \boldsymbol{W}$ through Variational Bayesian Networks (VBN) with *asymmetric* Bayesian weights of LoRA: $q_\theta(\boldsymbol{A}) = \mathcal{N}(\boldsymbol{A} | \boldsymbol{M}_A, \boldsymbol{\Omega}_A^2)$ and $q(\boldsymbol{B}) = \mathcal{N}(\boldsymbol{B} | \boldsymbol{0}, \boldsymbol{\Omega}_B^2)$ with $\boldsymbol{\Omega}_B \to \boldsymbol{0}^+$. The exact posterior $p(\boldsymbol{A} | \mathcal{D})$ is generally intractable, and the variational posterior $q_\theta(\boldsymbol{A})$ is learned through VI, which minimizes $\text{KL}[q_\theta(\boldsymbol{A}) \| p(\boldsymbol{A} | \mathcal{D})]$. The training objective is equivalent to minimizing the loss of variational free energy (VFE) with respect to variational parameters $\theta = \{\boldsymbol{M}_A, \boldsymbol{\Omega}_A^2\}$:

$$\mathcal{L}(\theta) = -\mathbb{E}_{q_\theta}\big[\log P(\mathcal{D}|\boldsymbol{A})\big] + \text{KL}\big[q_\theta(\boldsymbol{A}) \| p(\boldsymbol{A})\big], \quad (3)$$

where $p(\boldsymbol{A}) := \mathcal{N}(\boldsymbol{A} \mid \boldsymbol{0}, \sigma^2 \boldsymbol{I})$ is the prior of $\boldsymbol{A}$.

---

[1] https://github.com/WEIFZH/GPan-LoRA

## 2.2. GPs and Amortized VI

A GP $f(\cdot) \sim \mathcal{GP}(\mu, k)$ is fully specified by the mean function $\mu(\cdot)$ and the kernel function (covariance) $k(\cdot, \cdot')$. Let $\boldsymbol{x} \in \mathbb{R}^n$ denote a feature representation. We consider a $Q$-dimensional prediction head

$$\mathbf{f}(\boldsymbol{x}) = (f_1(\boldsymbol{x}), \dots, f_Q(\boldsymbol{x})), \tag{4}$$

modeled by *independent* GPs,

$$f_q(\boldsymbol{x}) \sim \mathcal{GP}(0, k(\boldsymbol{x}, \boldsymbol{x}')), \quad q = 1, \dots, Q, \tag{5}$$

sharing a common kernel $k$. The computational cost of exact GPs grows cubically with the number of training points, $O(N^3)$, making them unsuitable for use on large datasets. For more details, we direct readers to Williams & Rasmussen (2006).

For scalability, sparse variational GPs (Titsias, 2009) approximate each output associated with $M \ll N$ inducing variables $\mathbf{u}_q = f_q(\mathbf{Z})$, $\mathbf{Z} = \{\mathbf{z}_m\}_{m=1}^M$, with shared inputs across tasks. The variational posterior factorizes across $Q$ outputs: $q(\mathbf{u}) = \prod_{q=1}^Q \mathcal{N}(\boldsymbol{m}_q, \mathbf{S}_q)$.

Instead of optimizing $\theta := \{(\boldsymbol{m}_q, \mathbf{S}_q)\}_{q=1}^Q$ with VFE loss directly, we perform amortized VI (AVI) with a lightweight inference network $f_\phi^{\text{inf}}$ conditioned on hidden features:

$$\{(\boldsymbol{m}_q, \mathbf{S}_q)\}_{q=1}^Q = f_\phi^{\text{inf}}(\boldsymbol{x}), \quad q_\phi(\mathbf{u} \mid \boldsymbol{x}). \tag{6}$$

This enables variational parameters that depend on the input while still permitting efficient inference, and it also mitigates overconfidence, as discussed by Matias et al. (2024).

## 3. Related Work

### 3.1. Bayesian LoRA

Standard PEFT methods are deterministic and do not provide principled uncertainty estimates over task-specific adaptations. Recent work has begun to explicitly Bayesianize LoRA mechanisms for LLMs. Laplace-LoRA (LAP, Yang et al., 2023) introduces a probabilistic treatment of LoRA parameters, demonstrating that uncertainty-aware LoRA can improve robustness with calibration. Similarly, BLoB (Wang et al., 2024) is a variational approach that jointly optimizes the mean and covariance of LoRA parameters throughout the fine-tuning process, which leads to improved generalization and uncertainty estimation on both in-distribution and out-of-distribution data. Training-Free Bayesianization (TFB) (Shi et al., 2024) further develops BLoB by transforming existing off-the-shelf LoRA adapters into isotropic Gaussian distributions, offering maximally acceptable variance across deterministic LoRA backbones. C-LoRA (Rahmati et al., 2025) introduces an end-to-end Bayesian LoRA framework with contextual modules, allowing sample-dependent uncertainty.

## 3.2. GPs and UQ methods

GPs provide principled Bayesian function priors with well-calibrated uncertainty, and their close connection to NNs has long been established, with infinitely wide networks converging to GPs and kernel-based interpretations of deep models (Neal, 1996; Williams, 1996; Jacot et al., 2018). DKL further combines neural feature extractors with GPs to improve expressiveness, typically relying on sparse inducing approximations for scalability (Wilson et al., 2016a; Hensman et al., 2013). However, directly applying GP-based methods to LoRA models is non-trivial: placing GPs at multiple LoRA layers naturally induces a deep GP, leading to stacked uncertainty propagation and prohibitive inference complexity (Salimbeni & Deisenroth, 2017).

There are many other UQ methods, including MC dropout (MCD, Gal & Ghahramani, 2016) and deep ensemble (Deep-Ens, Lakshminarayanan et al., 2017). As shown in Table 1, existing UQ methods on LoRA models still face a fundamental trade-off between scalability and UQ quality, especially regarding the overfitting problem when distribution shifts exist. GPan-LoRA distinguishes itself by confining GP inference to the low-rank feature space, thereby achieving an effective balance that maintains both scalability and rigorous uncertainty for LLMs.

## 4. GPan-LoRA: Methodology

We now formally introduce GPan-LoRA for Bayesian low-rank adaptation. We extend the linear structure in (2) to a more general form of the hidden layer in a function space by **G**aussian **P**rocess **a**mortized **n**etworks (**GPan**):

$$\boldsymbol{h} = f_0(\boldsymbol{x}) + f_\Delta(\boldsymbol{x}), \tag{7}$$

where $f_0(\cdot)$ denotes the pretrained base model and is usually assumed to be $\boldsymbol{W}_0\boldsymbol{x}$ in LoRA. To facilitate uncertainty estimation, we employ GPs to represent the discrepancy between the outputs of the fine-tuned model and the pretrained model, $f_\Delta(\cdot) \sim \mathcal{GP}(\mu, k)$:

$$f_\Delta(\cdot) := g(\cdot) + \mu(\cdot), \tag{8}$$

where $g(\cdot) \sim \mathcal{GP}(0, k)$ is a *zero-mean* GP with covariance function $k(\cdot, \cdot')$, and $\mu(\cdot)$ is the mean function of $f_\Delta(\cdot)$.

To ensure scalability and efficiency, rather than directly solving $f_\Delta(\boldsymbol{x})$ in the high-dimensional space $\boldsymbol{x} \in \mathbb{R}^n$, we learn a low-rank adaptation of $g(\cdot)$ in the reduced space $\boldsymbol{A}\boldsymbol{x} \in \mathbb{R}^r$, formulated as a low-rank GP with amortized network, GPan-LoRA, as illustrated in Figure 2(c).

### 4.1. GP under low-rank feature maps

Let $g(\cdot) \sim \mathcal{GP}(0, k)$ be a GP on $\boldsymbol{x} \in \mathbb{R}^n$. Considering a low-dimensional linear transformation

$$\tau(\boldsymbol{x}) = \boldsymbol{A}\boldsymbol{x}, \quad \boldsymbol{A} \in \mathbb{R}^{r \times n}, \tag{9}$$

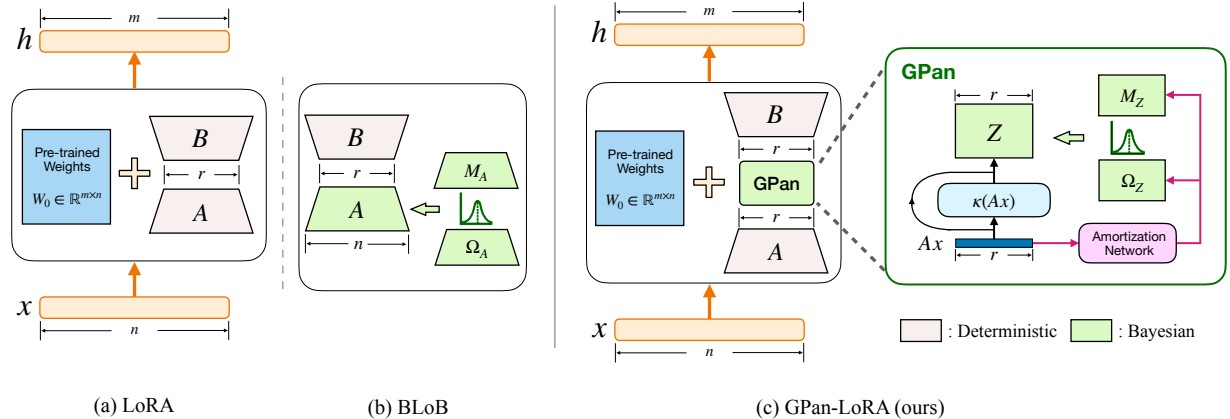

(a) LoRA    (b) BLoB    (c) GPan-LoRA (ours)

*Figure 2.* Module architectures of LoRA (left), BLoB (middle), and GPan-LoRA (right). While BLoB places Bayesian priors directly on LoRA factors, introducing Bayesian parameters that scale with the full input dimension $O(nr)$, GPan-LoRA confines Bayesian inference to the low-rank latent space, requiring only $O(lr^2)$ Bayesian variables via a low-rank GP amortized network.

we say that $g(\cdot)$ can be *exactly recast* through $\tau(\boldsymbol{x}) = \boldsymbol{Ax}$ if there exists a GP $\tilde{g}(\cdot) \sim \mathcal{GP}(0, \tilde{k})$ on $\mathbb{R}^r$ such that $\{g(\boldsymbol{x}) : \boldsymbol{x} \in \mathbb{R}^n\}$ and $\{\tilde{g}(\boldsymbol{Ax}) : \boldsymbol{Ax} \in \mathbb{R}^r\}$ have the same finite-dimensional distributions. Equivalently, the random function $g(\cdot)$ depends on $\boldsymbol{x}$ only through the projected coordinate $\tau(\boldsymbol{x}) = \boldsymbol{Ax}$.

**Theorem 4.1** (Exact GP recasting through linear transformations)**.** *The following statements are equivalent:*

*1) There exists a zero-mean GP $\tilde{g}(\cdot)$ on $\mathbb{R}^r$ such that $g(\boldsymbol{x}) \overset{\mathrm{d}}{=} \tilde{g}(\boldsymbol{Ax})$. 2) There exists a covariance function $\tilde{k}(\cdot, \cdot') : \mathbb{R}^r \times \mathbb{R}^r \to \mathbb{R}$ such that $k(\boldsymbol{x}, \boldsymbol{x}') = \tilde{k}(\boldsymbol{Ax}, \boldsymbol{Ax}')$.*

*Moreover, when these conditions hold, $\tilde{k}$ is positive semi-definite on the column space* range$(\boldsymbol{A}) \equiv \{\boldsymbol{Ax} : \boldsymbol{x} \in \mathbb{R}^n\} \subseteq \mathbb{R}^r$ *and therefore defines a valid GP.*

**Low-rank kernel.** While Theorem 4.1 holds for arbitrary kernels, we focus on stationary kernels in this work, which are the most commonly used choices in GPs. In particular, the squared exponential (SE) kernel $k_{\mathrm{SE}}(\boldsymbol{x}, \boldsymbol{x}')$ can be recast as an *isotropic* kernel $\pi_{\mathrm{SE}}$ with a low-rank lengthscale matrix $\Delta^{-1} = \boldsymbol{A}^T \boldsymbol{A}$ (de Souza et al., 2023) by applying a linear transformation $\boldsymbol{\tau}(\boldsymbol{x}) = \boldsymbol{Ax} : \mathbb{R}^n \to \mathbb{R}^r$ to $\boldsymbol{x} \in \mathbb{R}^n$:

$$k_{\mathrm{SE}}(\boldsymbol{x}, \boldsymbol{x}') = \sigma^2 \exp\left[-\frac{1}{2}(\boldsymbol{x} - \boldsymbol{x}')^\top \Delta^{-1}(\boldsymbol{x} - \boldsymbol{x}')\right] \quad (10)$$

$$= \sigma^2 \exp\left[-\frac{1}{2}\|\boldsymbol{Ax} - \boldsymbol{Ax}'\|^2\right] \quad (11)$$

$$= \pi_{\mathrm{SE}}\left(\boldsymbol{Ax}, \boldsymbol{Ax}'\right). \quad (12)$$

The corresponding $g(\cdot)$ is given by

$$\mathcal{GP}\left(0, \pi_{\mathrm{SE}}(\boldsymbol{Ax}, \boldsymbol{Ax}')\right) \Leftrightarrow \mathcal{GP}\left(0, k_{\mathrm{SE}}(\boldsymbol{x}, \boldsymbol{x}')\right). \quad (13)$$

### 4.2. Low-rank GP as a hidden layer of VBN

Simply merging low-rank adaptation with GPs does not *directly* yield a computationally efficient structure or an inference procedure that integrates well with LLMs, particularly in the context of a DGP model for maintaining the desired model capacity. To address the intensive computational complexity of DGPs, we approximate the GPs in each hidden layer with VBNs using the deep additive kernel.

**VBN approximation of $g(\cdot)$.** Let $\boldsymbol{u} \in \mathbb{R}^l$ represent the vector that collects the $l$ inducing points associated with each GP head. $g(\cdot)$ can be represented approximately using a sparsely additive induced kernel (Zhao et al., 2025):

$$g(\boldsymbol{Ax}) \approx \sum_{i=1}^{r} k(\boldsymbol{a}_i^\top \boldsymbol{x}, \boldsymbol{u}) \left[k(\boldsymbol{u}, \boldsymbol{u})\right]^{-1} g(\boldsymbol{u}) \quad (14)$$

$$= \sum_{i=1}^{r} k(\boldsymbol{a}_i^\top \boldsymbol{x}, \boldsymbol{u})[\boldsymbol{L}_{\boldsymbol{u}}^\top]^{-1} \boldsymbol{L}_{\boldsymbol{u}}^{-1} g(\boldsymbol{u}) \quad (15)$$

$$\triangleq \sum_{i=1}^{r} \boldsymbol{z}_i^\top \kappa(\boldsymbol{a}_i^\top \boldsymbol{x}, \boldsymbol{u}), \quad (16)$$

where $\boldsymbol{Ax} \in \mathbb{R}^r$ is the embedded low-rank feature, and $\boldsymbol{L}_{\boldsymbol{u}} \in \mathbb{R}^{l \times l}$ is the Cholesky decomposition of $k(\boldsymbol{u}, \boldsymbol{u})$.

Equation (15) can be decomposed into two components: $\kappa(\cdot) = \boldsymbol{L}_{\boldsymbol{u}}^{-1} k(\boldsymbol{u}, \cdot)$, which represents a nonlinear deterministic activation; and $\boldsymbol{z}_i = \boldsymbol{L}_{\boldsymbol{u}}^{-1} g(\boldsymbol{u}) \in \mathbb{R}^l$, which is an i.i.d. Gaussian random vector. Therefore, the induced kernel approximation of $g(\cdot)$ in (16) is mathematically equivalent to a hidden layer of VBN.

After vectorization and concatenation of $r$ independent GP heads, $f_\Delta(\boldsymbol{x})$ can ultimately be written in the matrix form

as a low-rank hidden GP layer:

$$f_\Delta(\boldsymbol{x}) := g(\boldsymbol{Ax}) + \mu(\boldsymbol{Ax}) \tag{17}$$

$$:= \boldsymbol{BZ}\kappa(\boldsymbol{Ax}) + \mu(\boldsymbol{Ax}), \tag{18}$$

where $\boldsymbol{A} \in \mathbb{R}^{r \times n}$ and $\boldsymbol{B} \in \mathbb{R}^{m \times r}$ are *deterministic* matrices, and $\boldsymbol{Z} \in \mathbb{R}^{r \times lr}$ is a *random* matrix, enabling efficient joint Bayesian inference.

**Bayesian residual mean $\mu(\cdot)$.** To enhance expressive capacity, we impose a priori on the mean function $\mu(\boldsymbol{Ax})$ and follow the asymmetric LoRA Bayesianization

$$\mu(\boldsymbol{x}) := \boldsymbol{B\Sigma Ax}, \tag{19}$$

where $\boldsymbol{A} \in \mathbb{R}^{r \times n}$ and $\boldsymbol{B} \in \mathbb{R}^{m \times r}$ are deterministic weight matrices shared with $g(\cdot)$, and $\boldsymbol{\Sigma} \in \mathbb{R}^{r \times r}$ is a Bayesian weight matrix. Equation (19) is equivalent to a low-rank SVD if we define $\boldsymbol{\Sigma} := \mathrm{diag}(\boldsymbol{\sigma})$ to be *diagonal* and positive semidefinite.

To save the cost on variational parameters, we provide a weight sharing strategy to inherently Bayesianize $\boldsymbol{\Sigma}$ through a pre-defined projection matrix $\boldsymbol{P}$:

$$f_\Delta(\boldsymbol{x}) = \boldsymbol{BZ}\big(\kappa(\boldsymbol{Ax}) + \boldsymbol{PAx}\big). \tag{20}$$

Therefore, $\mu(\boldsymbol{Ax})$ can be interpreted as a *residual* connection incorporated into the low-rank non-linearity $g(\boldsymbol{Ax})$.

### 4.3. Amortized Network and VI

Given the low-rank GP adaptation in (20), we perform variational inference (VI) and consider a variational family of mean-field Gaussian posteriors on random weights $q(\boldsymbol{Z}) = \mathcal{N}(\boldsymbol{Z} \mid \boldsymbol{M}_Z, \boldsymbol{\Omega}_Z^2)$, parameterized by $\theta := \{\boldsymbol{M}_Z, \boldsymbol{\Omega}_Z^2\}$.

As discussed in Section 2.2, amortized VI is designed to approximate posterior distributions using another input-dependent NN $f_\phi^{\mathrm{inf}}(\cdot)$. Given a low-rank feature $\boldsymbol{Ax}$, we use an amortization network to output variational parameters $\theta = f_\phi^{\mathrm{inf}}(\boldsymbol{Ax})$. Therefore, the amortized variational posterior is $q_\phi(\boldsymbol{Z} \mid \boldsymbol{x})$. The training objective is given by the VFE loss

$$\begin{aligned}\mathcal{L}(\boldsymbol{A}, \boldsymbol{B}, \phi) = &-\mathbb{E}_{q_\phi}\big[\log P(\mathcal{D} \mid \boldsymbol{A}, \boldsymbol{B}, \theta)\big] \\ &+ \mathrm{KL}\big[q_\phi(\theta|\boldsymbol{x})\|p_\phi(\theta)\big].\end{aligned} \tag{21}$$

During inference, we predict a new $\boldsymbol{x}^*$ by drawing $S$ samples from the variational distribution:

$$P(\boldsymbol{y}^* \mid \boldsymbol{x}^*, \mathcal{D}) \approx \frac{1}{S}\sum_{i=1}^{S} p(\boldsymbol{y}^* \mid \boldsymbol{x}^*, \boldsymbol{Z}_i), \tag{22}$$

where $\boldsymbol{Z}_i \sim q_\phi(\boldsymbol{Z}|\boldsymbol{x}_i)$ for $i = 1, \cdots, S$.

The amortized network $f_\phi^{\mathrm{inf}}(\cdot)$ shares the same neural feature extractor used to learn the latent feature $\boldsymbol{x}$, and therefore the input projection neural network becomes part of variational inference that helps to mitigate overfitting. Details can be found in Appendix A.5.

**Complexity.** Although Bayesian inference in GPan-LoRA is confined to a $r^2$-dimensional adaptation space, the computational complexity additionally depends on the number of inducing points $l$ used to approximate the underlying GP in (16), reflecting the standard trade-off in sparse GP approximations. In contrast, BLoB assigns Bayesian priors directly to LoRA factors, leading to a number of Bayesian parameters that grow with the full input dimension, i.e. $O(nr)$ where $n \gg lr$ in usual. C-LoRA similarly performs Bayesian inference in the $r^2$-dimensional space; however, it does not incorporate a function-space GP prior and therefore lacks the rigorous UQ guarantees provided by GPs.

Figure 2(c) shows the modular architecture of our GPan-LoRA, which requires only $O(lr^2)$ Bayesian variables through a low-rank GPan, making the computation, memory, and number of parameters comparable to the state-of-the-art Bayesian LoRA methods.

## 5. Experiments

In this section, we compare the proposed GPan-LoRA with existing LLM fine-tuning methods with UQ, and answer the following key research questions:

- How effective is GPan-LoRA in quantifying uncertainty, especially under distribution shifts? (Sec. 5.2 and Sec. 5.3)
- How do different components of GPan-LoRA affect the performance of UQ? (Sec. 5.4)
- Is GPan-LoRA aware of distribution shift? (Sec. 5.5)

### 5.1. Experimental Settings

**Base Models and Datasets.** To comprehensively demonstrate the UQ capability of our model in language understanding, we conduct experiments on two pretrained base models that are commonly used in the LLMs finetuning community: `LLaMA3.1-8B` (Grattafiori et al., 2024) and `Qwen3-8B` (Yang et al., 2025), and eight publicly available datasets: Winogrande-Small, Winigrande-Medium (Sakaguchi et al., 2021), ARC-Challenge, ARC-Easy (Clark et al., 2018), OBQA (Mihaylov et al., 2018), BoolQ (Clark et al., 2019), MMLU-Chem, and MMLU-Phy (Hendrycks et al., 2020). We focus our discussion based on the results on `LLaMA3.1-8B` as additional results on `Qwen3-8B` in Appendix C.4 show similar trends.

**Baselines.** We compare against several recent UQ methods with LoRA, including Monte-Carlo Dropout (**MCD**, Gal & Ghahramani, 2016), Deep Ensemble (**DeepEns**, Wang et al., 2023), Laplace-LoRA (**LAP**, Yang et al., 2023), Bayesian

LoRA by Backprop (**BLoB**, Wang et al., 2024), and Contextual LoRA (**C-LoRA**, Rahmati et al., 2025). We also include deterministic LoRA (**MLE**, Hu et al., 2022) for a comprehensive comparison.

**Fine-tuning Setups.** To ensure a fair and robust comparison, we partitioned the original training set, reserving 20% as a held-out validation set. We capped the training at 5,000 steps, ran validation every 500 steps, and used the checkpoint with the highest validation accuracy for the final evaluation. Detailed hyperparameter configurations are provided in Appendix B.2.

**Metrics.** The datasets consist of either binary or multiple-choice QA problems. We adopt three commonly used metrics for UQ evaluation (Wang et al., 2024; Rahmati et al., 2025) on these datasets: accuracy (ACC), expected calibrated error (ECE), and negative log-likelihood (NLL).

## 5.2. Performance Comparison

**In-Distribution.** We first evaluate GPan-LoRA on in-distribution (ID) datasets as reported in Table 2. Overall, GPan-LoRA achieves competitive accuracy while providing strong uncertainty estimates.

*Predictive Accuracy.* On ID tasks, GPan-LoRA maintains comparable accuracy to deterministic fine-tuning (MLE) and other UQ baselines, indicating that our Bayesianization does not compromise task adaptation. While no single method uniformly dominates across all datasets, GPan-LoRA remains within a competitive range and preserves the base model's generative capability.

*Uncertainty Calibration.* In terms of calibration, GPan-LoRA is consistently strong and achieves the best or second-best ECE across all datasets, outperforming recent Bayesian methods on these benchmarks. For NLL, although LAP and C-LoRA are highly competitive, GPan-LoRA attains the lowest NLL on ARC-C/E and OBQA, and is comparable to the best method on other datasets. These results suggest that GPan-LoRA can match strong Bayesian baselines on ID performance while improving calibration on multiple tasks.

**Out-of-Distribution.** We further assess robustness under distribution shift, where models trained on OBQA are evaluated directly on unseen targets. We group targets into *small* and *large* shifts as illustrated in the right part of Table 2.

*Small Distribution Shifts.* On mild shifts (ARC-C/E), although GPan-LoRA shows lower ACC compared to LAP, it does not compromise calibration quality. In fact, GPan-LoRA remains a top-tier performer in uncertainty quantification: it achieves the best and second-best ECE on ARC-E and ARC-C, respectively. This indicates that while GPan-LoRA may be less aggressive in fitting the source domain's

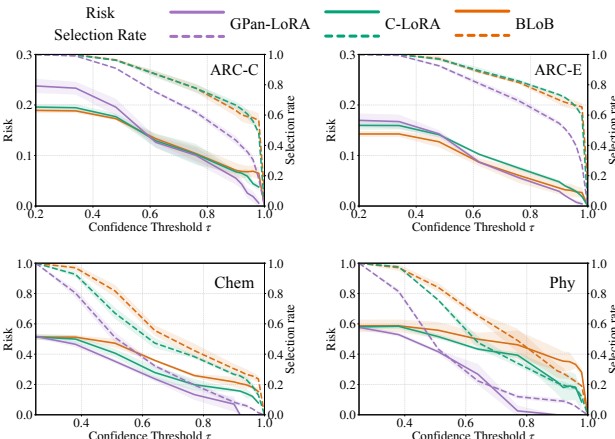

*Figure 3.* Risk and selection rate vs. confidence threshold $\tau$ under distribution shift. Solid: risk; dashed: selection rate; shading: $\pm 1$ std over seeds.

accuracy peaks, it successfully preserves reliable probabilistic estimates during near-domain transfer. The illustrative example in Figure 1 also confirms this: GPan-LoRA may have a more conservative prediction error but better uncertainty calibration. Further discussion can be found in Section 5.3 and Appendix B.1.

*Large Distribution Shifts.* Under severe shifts to specialized STEM domains (Chem/Phy), the robustness of GPan-LoRA becomes the decisive factor. It achieves *state-of-the-art* UQ quality, recording the lowest ECE on both Chem and Phy (11.31/13.29) and the lowest NLL (1.19/1.25). Crucially, it outperforms the strongest baselines by a large margin: compared to LAP (ECE 14.59/14.00) and BLoB (ECE 24.45/32.16), GPan-LoRA reduces calibration error significantly. We hypothesize that this consistent superiority stems from GPan-LoRA's ability to better capture epistemic uncertainty when the input distribution deviates substantially from the training domain. Detailed discussion can be found in Section 5.5.

**Last Layer Analysis.** GPan-LoRA is demonstrated achieving more robust performances under distribution shift with comparable computation, memory, and number of parameters to the other state-of-the-art approaches. For budget-constrained application scenarios, it may be desirable to adopt a resource-light option without sacrificing too much performance. Motivated by previous work suggesting that incorporating uncertainty only in the last layer can preserve most of the advantages of full Bayesian treatment while significantly reducing computational cost (Wilson et al., 2016a; Harrison et al., 2024), we have investigated a lightweight variant of GPan that Bayesianizes only the last LoRA layer, denoted as **LL GPan-LoRA**. We evaluate its UQ performance under the same experimental setup as the full GPan-LoRA model, with results summarized in Table 2. As expected, restricting Bayesianization to the

*Table 2.* **Performance of different uncertainty estimation methods applied to LoRA on** `LLaMA3.1-8B` **pre-trained weights.** Accuracy (**ACC**) and Expected Calibration Error (**ECE**) are reported in percentages, where "↑" and "↓" indicate that higher and lower values are preferred, respectively. The left six columns correspond to the in-distribution setting, while the right four columns report performance under distribution shift. Best and second-best ECE and NLL results are highlighted in **boldface** and underline, respectively.

| Metric | Method | In-Distribution Datasets | | | | | | Out-of-Distribution Datasets (OBQA→X) | | | |
| | | | | | | | | Small Dist. Shift | | Large Dist. Shift | |
| | | WG-S | ARC-C | ARC-E | WG-M | OBQA | BoolQ | ARC-C | ARC-E | Chem | Phy |
| ACC (↑) | MLE | $76.18_{\pm1.45}$ | $79.39_{\pm0.67}$ | $91.43_{\pm1.27}$ | $81.27_{\pm1.70}$ | $86.26_{\pm0.41}$ | $89.60_{\pm0.80}$ | $80.40_{\pm1.35}$ | $85.15_{\pm0.90}$ | $48.33_{\pm1.52}$ | $39.33_{\pm2.88}$ |
| | MCD | $75.50_{\pm0.71}$ | $80.85_{\pm1.27}$ | $90.25_{\pm0.36}$ | $81.90_{\pm0.53}$ | $86.66_{\pm1.33}$ | $89.43_{\pm0.67}$ | $80.51_{\pm0.51}$ | $85.91_{\pm0.63}$ | $48.00_{\pm2.64}$ | $42.00_{\pm4.00}$ |
| | DeepEns | $76.74_{\pm1.07}$ | $80.91_{\pm0.71}$ | $91.66_{\pm0.26}$ | $81.85_{\pm1.49}$ | $87.00_{\pm0.20}$ | $89.88_{\pm0.24}$ | $81.30_{\pm0.39}$ | $85.97_{\pm0.44}$ | $44.00_{\pm3.46}$ | $45.66_{\pm1.52}$ |
| | LAP | $74.55_{\pm0.51}$ | $81.47_{\pm0.42}$ | $91.48_{\pm0.36}$ | $85.73_{\pm0.34}$ | $80.40_{\pm0.27}$ | $87.87_{\pm0.36}$ | $80.68_{\pm1.02}$ | $88.01_{\pm0.71}$ | $47.00_{\pm2.00}$ | $44.44_{\pm0.57}$ |
| | BLoB | $72.12_{\pm0.43}$ | $79.73_{\pm1.75}$ | $90.84_{\pm0.63}$ | $79.40_{\pm0.31}$ | $86.86_{\pm0.41}$ | $89.76_{\pm0.16}$ | $81.64_{\pm1.08}$ | $86.67_{\pm1.78}$ | $49.00_{\pm1.00}$ | $43.00_{\pm4.58}$ |
| | C-LoRA | $73.76_{\pm2.28}$ | $80.40_{\pm0.67}$ | $89.61_{\pm0.46}$ | $76.26_{\pm2.06}$ | $87.20_{\pm0.34}$ | $89.54_{\pm0.24}$ | $80.74_{\pm0.67}$ | $83.68_{\pm0.44}$ | $49.00_{\pm1.73}$ | $42.66_{\pm2.08}$ |
| | GPan-LoRA | $72.41_{\pm0.24}$ | $80.06_{\pm1.01}$ | $91.84_{\pm0.61}$ | $79.82_{\pm0.44}$ | $86.66_{\pm0.94}$ | $87.20_{\pm3.29}$ | $75.78_{\pm1.36}$ | $82.74_{\pm1.23}$ | $48.00_{\pm2.00}$ | $42.33_{\pm0.57}$ |
| | LL GPan-LoRA | $75.60_{\pm0.09}$ | $79.27_{\pm2.24}$ | $90.72_{\pm0.86}$ | $78.63_{\pm0.55}$ | $86.13_{\pm0.83}$ | $88.97_{\pm0.30}$ | $79.84_{\pm1.52}$ | $84.03_{\pm0.56}$ | $49.66_{\pm2.08}$ | $39.33_{\pm4.50}$ |
| ECE (↓) | MLE | $22.43_{\pm0.94}$ | $17.01_{\pm1.80}$ | $6.33_{\pm2.21}$ | $13.43_{\pm2.62}$ | $6.93_{\pm2.84}$ | $4.24_{\pm1.73}$ | $16.16_{\pm1.34}$ | $11.30_{\pm0.34}$ | $32.60_{\pm2.47}$ | $41.61_{\pm1.17}$ |
| | MCD | $23.36_{\pm0.77}$ | $15.42_{\pm3.50}$ | $4.55_{\pm2.60}$ | $15.24_{\pm1.52}$ | $9.01_{\pm0.59}$ | $5.51_{\pm1.17}$ | $11.74_{\pm2.81}$ | $8.48_{\pm1.28}$ | $24.33_{\pm5.09}$ | $32.07_{\pm6.98}$ |
| | DeepEns | $21.80_{\pm1.12}$ | $17.50_{\pm1.07}$ | $6.93_{\pm0.64}$ | $14.07_{\pm2.04}$ | $9.98_{\pm0.65}$ | $6.05_{\pm0.17}$ | $9.02_{\pm0.09}$ | $9.69_{\pm1.39}$ | $25.37_{\pm3.32}$ | $28.45_{\pm1.46}$ |
| | LAP | $\mathbf{7.78_{\pm1.92}}$ | $5.81_{\pm0.96}$ | $\underline{3.50_{\pm0.65}}$ | $\mathbf{4.31_{\pm0.26}}$ | $3.67_{\pm0.21}$ | $1.75_{\pm0.07}$ | $\mathbf{5.01_{\pm0.93}}$ | $\underline{3.43_{\pm1.11}}$ | $14.59_{\pm2.41}$ | $14.00_{\pm1.87}$ |
| | BLoB | $17.29_{\pm0.59}$ | $8.68_{\pm4.33}$ | $4.36_{\pm2.45}$ | $13.66_{\pm1.75}$ | $5.78_{\pm0.61}$ | $7.28_{\pm0.37}$ | $8.26_{\pm0.94}$ | $5.37_{\pm1.78}$ | $24.45_{\pm1.20}$ | $32.16_{\pm4.57}$ |
| | C-LoRA | $14.50_{\pm1.43}$ | $5.68_{\pm2.06}$ | $4.16_{\pm0.84}$ | $9.91_{\pm1.67}$ | $5.35_{\pm0.34}$ | $4.35_{\pm0.16}$ | $8.16_{\pm1.74}$ | $7.20_{\pm0.15}$ | $19.39_{\pm2.22}$ | $25.68_{\pm2.98}$ |
| | GPan-LoRA | $13.39_{\pm2.67}$ | $\mathbf{4.60_{\pm0.88}}$ | $\mathbf{2.36_{\pm0.85}}$ | $\underline{6.85_{\pm1.44}}$ | $\mathbf{3.12_{\pm0.64}}$ | $\mathbf{1.45_{\pm0.62}}$ | $\underline{6.46_{\pm0.87}}$ | $\mathbf{3.31_{\pm0.37}}$ | $\mathbf{11.31_{\pm2.75}}$ | $\mathbf{13.29_{\pm1.20}}$ |
| | LL GPan-LoRA | $15.72_{\pm3.18}$ | $7.93_{\pm2.27}$ | $3.87_{\pm0.54}$ | $8.25_{\pm2.49}$ | $6.07_{\pm1.70}$ | $2.13_{\pm0.20}$ | $8.01_{\pm0.56}$ | $3.67_{\pm0.66}$ | $\underline{14.37_{\pm2.24}}$ | $\underline{13.67_{\pm3.00}}$ |
| NLL (↓) | MLE | $1.92_{\pm0.18}$ | $1.35_{\pm0.46}$ | $0.43_{\pm0.12}$ | $0.63_{\pm0.10}$ | $0.41_{\pm0.03}$ | $0.28_{\pm0.02}$ | $1.10_{\pm0.03}$ | $0.78_{\pm0.05}$ | $1.96_{\pm0.11}$ | $2.38_{\pm0.17}$ |
| | MCD | $2.28_{\pm0.13}$ | $0.85_{\pm0.12}$ | $0.35_{\pm0.12}$ | $0.92_{\pm0.23}$ | $0.50_{\pm0.06}$ | $0.30_{\pm0.02}$ | $0.74_{\pm0.13}$ | $0.45_{\pm0.09}$ | $1.47_{\pm0.20}$ | $1.72_{\pm0.32}$ |
| | DeepEns | $1.78_{\pm0.15}$ | $1.51_{\pm0.03}$ | $0.48_{\pm0.14}$ | $0.72_{\pm0.12}$ | $0.57_{\pm0.10}$ | $0.31_{\pm0.01}$ | $0.59_{\pm0.01}$ | $0.48_{\pm0.01}$ | $1.26_{\pm0.04}$ | $1.36_{\pm0.02}$ |
| | LAP | $\mathbf{0.56_{\pm0.02}}$ | $\underline{0.51_{\pm0.01}}$ | $0.26_{\pm0.01}$ | $\mathbf{0.45_{\pm0.01}}$ | $0.38_{\pm0.01}$ | $0.27_{\pm0.01}$ | $\mathbf{0.50_{\pm0.02}}$ | $\underline{0.35_{\pm0.02}}$ | $1.28_{\pm0.03}$ | $\underline{1.29_{\pm0.01}}$ |
| | BLoB | $0.94_{\pm0.01}$ | $0.65_{\pm0.04}$ | $0.36_{\pm0.07}$ | $0.72_{\pm0.03}$ | $0.46_{\pm0.04}$ | $0.41_{\pm0.01}$ | $0.68_{\pm0.27}$ | $0.44_{\pm0.05}$ | $1.61_{\pm0.07}$ | $1.91_{\pm0.10}$ |
| | C-LoRA | $0.70_{\pm0.06}$ | $0.56_{\pm0.01}$ | $0.29_{\pm0.01}$ | $0.55_{\pm0.03}$ | $0.38_{\pm0.01}$ | $0.27_{\pm0.01}$ | $0.60_{\pm0.04}$ | $0.44_{\pm0.01}$ | $1.35_{\pm0.06}$ | $1.51_{\pm0.06}$ |
| | GPan-LoRA | $\underline{0.69_{\pm0.08}}$ | $\mathbf{0.51_{\pm0.02}}$ | $\mathbf{0.24_{\pm0.01}}$ | $\underline{0.48_{\pm0.01}}$ | $\mathbf{0.37_{\pm0.02}}$ | $0.30_{\pm0.06}$ | $0.60_{\pm0.03}$ | $0.46_{\pm0.01}$ | $\mathbf{1.19_{\pm0.01}}$ | $\mathbf{1.25_{\pm0.01}}$ |
| | LL GPan-LoRA | $0.78_{\pm0.14}$ | $0.58_{\pm0.05}$ | $0.28_{\pm0.01}$ | $0.51_{\pm0.03}$ | $0.39_{\pm0.01}$ | $\mathbf{0.26_{\pm0.01}}$ | $\underline{0.56_{\pm0.03}}$ | $\mathbf{0.38_{\pm0.02}}$ | $\underline{1.25_{\pm0.05}}$ | $\mathbf{1.25_{\pm0.06}}$ |

last layer leads to moderate performance degradation across most datasets. Nevertheless, this efficient variant remains competitive with existing baselines and notably achieves strong performance under large distribution shifts, where it consistently outperforms prior methods. These results suggest that last-layer Bayesianization offers a favorable trade-off between computational efficiency and robust uncertainty estimation. Detailed efficiency analysis can be found in Appendix C.1.

### 5.3. Selective Prediction Under Distribution Shift

In this section, we study selective prediction under distribution shift by allowing the model to abstain when its confidence is below a threshold $\tau$. We focus on end-to-end Bayesian LoRA methods that are trained and deployed under comparable computational budgets. For each input $x$, the model outputs a predictive distribution over $C$ answer options. We define confidence as $\max_{c\in[C]} p_\theta(y = c \mid x)$. The model selects an example if its confidence exceeds $\tau$, yielding the **selection rate**: $\mathrm{Sel}(\tau) = \mathbb{P}(\max_c p_\theta(y = c \mid x) \geq \tau)$, and the corresponding **risk** on the selected subset: $\mathrm{Risk}(\tau) = \mathbb{E}[\mathbb{I}(\hat{y}(x) \neq y) \mid \max_c p_\theta(y = c \mid x) \geq \tau]$. We sweep $\tau$ from $1/C$ to 1 and report both $\mathrm{Risk}(\tau)$ and $\mathrm{Sel}(\tau)$, as shown in Figure 3.

**Small Distribution shift: reliability emerges at high con-**

**fidence.** Under the small distribution shift regime (Table 2), GPan-LoRA does not yield the best overall accuracy. Nonetheless, Figures 3(a-b) show that as $\tau$ increases, GPan-LoRA's risk decreases much faster than the baselines, and in the high-confidence regime it consistently achieves the lowest risk. This indicates that even when average accuracy is not dominant, GPan-LoRA better identifies instances where it is likely to be correct and abstains on the rest, making it preferable when the application requires *extremely reliable* predictions.

**Large Distribution shift: lower risk without over-confidence.** Under a large distribution shift as shown in Figures 3(c-d), GPan-LoRA's risk is consistently below both baselines across a wide range of $\tau$. Importantly, when viewed together with the selection-rate curves, GPan-LoRA does not maintain an artificially high selection rate at the expense of errors; instead, it reduces the selection rate appropriately as $\tau$ increases. This behavior suggests that GPan-LoRA is less over-confident under severe shift: it assigns lower confidence to unfamiliar or hard examples, abstains more often, and consequently maintains lower selective risk than baselines. At the same time, when operating at high-confidence thresholds, GPan-LoRA achieves higher prediction accuracy, indicating that its confidence is well aligned with true predictive reliability.

Overall, these results highlight that selective prediction pro-

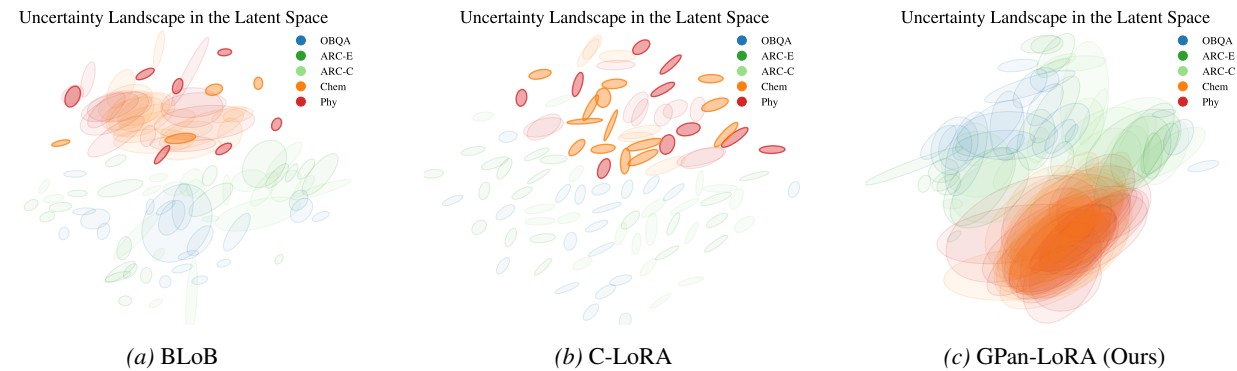

Uncertainty Landscape in the Latent Space    Uncertainty Landscape in the Latent Space    Uncertainty Landscape in the Latent Space

*(a)* BLoB                          *(b)* C-LoRA                          *(c)* GPan-LoRA (Ours)

*Figure 4.* t-SNE visualization of the latent embeddings under distribution shift.

*Table 3.* Ablation: performance of GPan-LoRA and its variants under distribution shift.

| Metric | Method | In-Dist. | Out-of-Dist. (OBQA→X) | | | |
|---|---|---|---|---|---|---|
| | | | *Small Dist. Shift* | | *Large Dist. Shift* | |
| | | OBQA | ARC-C | ARC-E | Chem | Phy |
| ACC ↑ | GPan-LoRA | $86.66_{\pm0.94}$ | $75.78_{\pm1.36}$ | $82.74_{\pm1.23}$ | $48.00_{\pm2.00}$ | $42.33_{\pm0.51}$ |
| | w/o AN | $87.27_{\pm0.83}$ | $80.96_{\pm0.19}$ | $85.79_{\pm1.23}$ | $42.00_{\pm3.00}$ | $41.66_{\pm3.21}$ |
| | w/o WS | $84.73_{\pm0.80}$ | $75.78_{\pm1.18}$ | $80.16_{\pm0.61}$ | $44.33_{\pm1.52}$ | $38.66_{\pm3.21}$ |
| ECE ↓ | GPan-LoRA | $\mathbf{3.12}_{\pm\mathbf{0.64}}$ | $6.46_{\pm0.87}$ | $\mathbf{3.31}_{\pm\mathbf{0.37}}$ | $\mathbf{11.31}_{\pm\mathbf{2.75}}$ | $\mathbf{13.29}_{\pm\mathbf{1.20}}$ |
| | w/o AN | $3.71_{\pm0.97}$ | $\mathbf{6.35}_{\pm\mathbf{0.38}}$ | $4.14_{\pm1.03}$ | $22.60_{\pm0.94}$ | $26.15_{\pm3.49}$ |
| | w/o WS | $3.70_{\pm1.00}$ | $7.92_{\pm0.23}$ | $4.62_{\pm0.59}$ | $19.67_{\pm2.51}$ | $22.52_{\pm0.91}$ |
| NLL ↓ | GPan-LoRA | $0.37_{\pm0.02}$ | $0.60_{\pm0.03}$ | $0.46_{\pm0.01}$ | $\mathbf{1.19}_{\pm\mathbf{0.01}}$ | $\mathbf{1.25}_{\pm\mathbf{0.01}}$ |
| | w/o AN | $\mathbf{0.36}_{\pm\mathbf{0.01}}$ | $\mathbf{0.57}_{\pm\mathbf{0.03}}$ | $\mathbf{0.39}_{\pm\mathbf{0.04}}$ | $1.37_{\pm0.02}$ | $1.50_{\pm0.05}$ |
| | w/o WS | $0.39_{\pm0.01}$ | $0.66_{\pm0.01}$ | $0.53_{\pm0.02}$ | $1.34_{\pm0.04}$ | $1.39_{\pm0.03}$ |

vides a complementary view to average accuracy: GPan-LoRA is particularly advantageous in high-reliability operating regions and under large distribution shift, where well-calibrated abstention is crucial.

### 5.4. Ablation Study

We investigate the contributions of the amortized network and weight sharing in GPan-LoRA by ablating each component individually, resulting in two variants: w/o amortized network (w/o AN) and w/o weight sharing (w/o WS), as shown in Table 3.

**W/o amortized network.** In this variant, we remove the amortized inference mechanism introduced in Section 4.3, the variational parameters are optimized independently without amortization. While this variant achieves comparable or slightly higher in-distribution accuracy, it consistently exhibits worse uncertainty calibration under both small and large distribution shifts. This ablation suggests that amortized inference is critical for learning input-aware variational posteriors, enabling GPan-LoRA to maintain reliable uncertainty estimates when encountering unseen inputs.

**W/o weight sharing.** For the w/o WS variant, the mean function and kernel function in Equation (17) no longer share parameters, leading to a larger number of independent Bayesian parameters. As shown in Table 3, this modification results in consistent performance degradation across both

accuracy and uncertainty metrics, especially under large distribution shifts. The increased ECE and NLL indicate that removing weight sharing negatively affects posterior learning, likely due to over-parameterization and reduced statistical efficiency.

### 5.5. Visualization of Uncertainty Landscape

To further investigate how GPan-LoRA quantifies uncertainty under varying degrees of distribution shift, we visualize the variance of latent space embeddings. Following the experimental protocol for uncertainty visualization, we extract the output embeddings from the final transformer layer for two variational inference based baselines and our proposed GPan-LoRA. Detailed visualization setup is provided in Appendix B.3.

**Results and Observations.** As illustrated in Figure 4, our comparative analysis yields several key insights:

*Overconfidence in Baselines.* The baseline models exhibit varying degrees of overconfidence when encountering OOD data. Specifically, under distribution shifts, these baselines yield tightly clustered embeddings (highlighted with bold border in the figure). This indicates a failure to capture epistemic uncertainty, as the models provide low-variance predictions for samples far from the training distribution.

*Robust Uncertainty Estimation in GPan-LoRA.* In contrast, GPan-LoRA demonstrates a superior ability to remain "cautious." As shown in Figure 4(c), GPan-LoRA maintains significantly higher uncertainty (manifested as larger, more dispersed ellipses) for OOD MMLU samples compared to the ID OBQA samples. This behavior aligns with the motivating regression example in Figure 1, confirming that GPan-LoRA successfully replicates the desirable property of GPs of expanding uncertainty estimates in OOD regions within the representation space induced by LLMs.

*Sensitivity to Shift Magnitude.* Our model also distinguishes between different levels of shift: while the uncertainty for ARC datasets is moderately increased, the expansion is most pronounced for the MMLU datasets. Combined with

discussion in Section 5.3, this confirms that GPan-LoRA effectively leverages its Bayesian framework to mitigate the risk of overconfident mispredictions in complex, unseen domains.

## 6. Conclusion

In this work, we presented GPan-LoRA, a Bayesian LoRA framework via Gaussian Process amortized networks, that brings principled uncertainty quantification to parameter-efficient fine-tuning of LLMs. By reformulating sparse GPs as low-rank Bayesian adapters and learning their variational posteriors through amortized networks, GPan-LoRA achieves strong uncertainty quality while operating entirely in a compact $r^2$-dimensional adaptation space. Experiments demonstrate that this design yields well-calibrated uncertainty and mitigates overconfidence under distribution shift without sacrificing efficiency, positioning GPan-LoRA as a practical and theoretically grounded solution for uncertainty-aware LLM adaptation.

## Acknowledgements

This work was supported in part by the U.S. National Science Foundation (NSF) grants SHF-2215573, and IIS-2212419. The work of W. Zhao and C. Tian was supported in part by the NSF via grants DMS-2312173 and ECCS-2433631. Portions of this research were conducted with the advanced computing resources provided by Texas A&M High Performance Research Computing.

## Impact Statement

This paper presents work whose goal is to advance the fields of Gaussian Processes, Bayesian Learning, and Large Language Models. There are many potential societal consequences of our work, especially in tackling challenges in achieving trustworthy AI/ML.

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

# A. Background: Gaussian Process and Variational Inference

## A.1. Exact Gaussian Processes

A Gaussian Process (GP) defines a distribution over real-valued functions $f : \mathbb{R}^D \to \mathbb{R}$ such that any finite collection of function values follows a joint Gaussian distribution. Formally,

$$f(\cdot) \sim \mathcal{GP}\big(m(\cdot), k(\cdot, \cdot)\big), \tag{23}$$

where $m(\cdot)$ is the mean function (typically set to zero) and $k(\cdot, \cdot)$ is a positive-definite kernel.

Given training inputs $\mathbf{X} = \{x_i\}_{i=1}^N$ and noisy observations

$$y_i = f(x_i) + \varepsilon_i, \quad \varepsilon_i \sim \mathcal{N}(0, \sigma^2), \tag{24}$$

the prior over latent function values $\mathbf{f} = f(\mathbf{X})$ is

$$p(\mathbf{f}) = \mathcal{N}\big(\mathbf{0}, \mathbf{K_{XX}}\big), \tag{25}$$

where $(\mathbf{K_{XX}})_{ij} = k(x_i, x_j)$. The marginal likelihood is

$$p(\mathbf{y}) = \mathcal{N}\big(\mathbf{0}, \mathbf{K_{XX}} + \sigma^2 \mathbf{I}\big). \tag{26}$$

For a test input $x_*$, let $f_* = f(x_*)$. The joint prior over $(\mathbf{f}, f_*)$ is

$$\begin{bmatrix} \mathbf{f} \\ f_* \end{bmatrix} \sim \mathcal{N}\left(\mathbf{0}, \begin{bmatrix} \mathbf{K_{XX}} & \mathbf{k_{X*}} \\ \mathbf{k_{*X}} & k_{**} \end{bmatrix}\right), \tag{27}$$

where $\mathbf{k_{X*}} = k(\mathbf{X}, x_*)$. Conditioning on observations $\mathbf{y}$ yields the exact GP posterior:

$$p(f_* \mid \mathbf{y}) = \mathcal{N}(\mu_*, \sigma_*^2), \tag{28}$$
$$\mu_* = \mathbf{k_{*X}}(\mathbf{K_{XX}} + \sigma^2 \mathbf{I})^{-1}\mathbf{y}, \tag{29}$$
$$\sigma_*^2 = k_{**} - \mathbf{k_{*X}}(\mathbf{K_{XX}} + \sigma^2 \mathbf{I})^{-1}\mathbf{k_{X*}}. \tag{30}$$

While exact GPs provide closed-form posterior inference and well-calibrated uncertainty estimates, their computational complexity scales as $\mathcal{O}(N^3)$ in training and $\mathcal{O}(N^2)$ in prediction, which limits applicability to large-scale datasets.

## A.2. Sparse Variational Gaussian Processes with Inducing Points

Sparse variational GP (SVGP) methods alleviate the cubic complexity by introducing a set of $M \ll N$ inducing points $\mathbf{Z} = \{z_m\}_{m=1}^M$ with corresponding inducing variables $\mathbf{u} = f(\mathbf{Z})$. Under the GP prior,

$$p(\mathbf{u}) = \mathcal{N}\big(\mathbf{0}, \mathbf{K_{ZZ}}\big), \quad p(\mathbf{f} \mid \mathbf{u}) = \mathcal{N}\big(\mathbf{K_{XZ}}\mathbf{K_{ZZ}}^{-1}\mathbf{u}, \mathbf{K_{XX}} - \mathbf{Q_{XX}}\big), \tag{31}$$

where $\mathbf{Q_{XX}} = \mathbf{K_{XZ}}\mathbf{K_{ZZ}}^{-1}\mathbf{K_{ZX}}$.

Variational inference is applied by introducing a tractable variational distribution over inducing variables,

$$q(\mathbf{u}) = \mathcal{N}(\mathbf{m}, \mathbf{S}), \tag{32}$$

which induces an approximate posterior over function values

$$q(\mathbf{f}) = \int p(\mathbf{f} \mid \mathbf{u}) q(\mathbf{u}) \mathrm{d}\mathbf{u}. \tag{33}$$

The variational parameters $(\mathbf{m}, \mathbf{S})$ are optimized by maximizing the evidence lower bound (ELBO):

$$\mathcal{L}_{\mathrm{ELBO}} = \mathbb{E}_{q(\mathbf{f})}[\log p(\mathbf{y} \mid \mathbf{f})] - \mathrm{KL}\big(q(\mathbf{u}) \,\|\, p(\mathbf{u})\big). \tag{34}$$

Sparse variational GPs reduce training complexity to $\mathcal{O}(NM^2)$ and prediction to $\mathcal{O}(M^2)$, while retaining a principled Bayesian treatment of uncertainty.

### A.3. From sparse GP approximations to variational Bayesian networks

Although SVGP reduces the complexity from $O(N^3)$ to $O(NM^2)$, it still misaligns the training and inference techniques in modern deep neural networks. To further improve the computation, we provide a functional interpretation of sparse GP approximations that reveals their close connection to variational Bayesian neural networks (VBNs) with structured, kernel-induced features.

Consider a latent function $g(\cdot)$ endowed with a GP prior. In sparse variational GPs, a set of inducing inputs $\boldsymbol{u} \in \mathbb{R}^M$ is introduced, together with inducing variables $g(\boldsymbol{u})$. Conditioned on $g(\boldsymbol{u})$, the GP prior admits the approximation

$$g(\boldsymbol{x}) \approx k(\boldsymbol{x}, \boldsymbol{u})\, k(\boldsymbol{u}, \boldsymbol{u})^{-1} g(\boldsymbol{u}), \tag{35}$$

which corresponds to projecting the function onto the subspace spanned by kernel evaluations at inducing points. This approximation becomes exact as $M \to \infty$ under mild conditions.

We now specialize to the setting where the GP is defined over a low-rank linear projection of the input, i.e.

$$g(\boldsymbol{x}) = g(\boldsymbol{A}\boldsymbol{x}),$$

with $\boldsymbol{A} = [\boldsymbol{a}_1, \ldots, \boldsymbol{a}_r]^\top \in \mathbb{R}^{r \times n}$. Assuming an additive kernel over projected coordinates, the kernel admits the decomposition

$$k(\boldsymbol{A}\boldsymbol{x}, \boldsymbol{u}) = \sum_{i=1}^{r} k(\boldsymbol{a}_i^\top \boldsymbol{x}, \boldsymbol{u}). \tag{36}$$

Applying the approximation in (35) yields

$$g(\boldsymbol{A}\boldsymbol{x}) \approx k(\boldsymbol{A}\boldsymbol{x}, \boldsymbol{u})\, k(\boldsymbol{u}, \boldsymbol{u})^{-1} g(\boldsymbol{u}) \tag{37}$$

$$= \sum_{i=1}^{r} k(\boldsymbol{a}_i^\top \boldsymbol{x}, \boldsymbol{u})\, k(\boldsymbol{u}, \boldsymbol{u})^{-1} g(\boldsymbol{u}). \tag{38}$$

Let $k(\boldsymbol{u}, \boldsymbol{u}) = \boldsymbol{L}_{\boldsymbol{u}} \boldsymbol{L}_{\boldsymbol{u}}^\top$ denote the Cholesky factorization of the inducing kernel matrix. Then

$$g(\boldsymbol{A}\boldsymbol{x}) \approx \sum_{i=1}^{r} k(\boldsymbol{a}_i^\top \boldsymbol{x}, \boldsymbol{u})\, [\boldsymbol{L}_{\boldsymbol{u}}^\top]^{-1} \boldsymbol{L}_{\boldsymbol{u}}^{-1} g(\boldsymbol{u}) \tag{39}$$

$$\triangleq \sum_{i=1}^{r} \boldsymbol{z}_i^\top \kappa(\boldsymbol{a}_i^\top \boldsymbol{x}, \boldsymbol{u}), \tag{40}$$

where we define

$$\kappa(\boldsymbol{a}_i^\top \boldsymbol{x}, \boldsymbol{u}) := k(\boldsymbol{a}_i^\top \boldsymbol{x}, \boldsymbol{u})\, [\boldsymbol{L}_{\boldsymbol{u}}^\top]^{-1}, \qquad \boldsymbol{z}_i := \boldsymbol{L}_{\boldsymbol{u}}^{-1} g(\boldsymbol{u}).$$

Under the GP prior, $g(\boldsymbol{u}) \sim \mathcal{N}(\boldsymbol{0}, k(\boldsymbol{u}, \boldsymbol{u}))$, which implies

$$\boldsymbol{z}_i \sim \mathcal{N}(\boldsymbol{0}, \boldsymbol{I}),$$

and hence each $\boldsymbol{z}_i$ can be interpreted as a standard Gaussian latent weight vector. Consequently, the sparse GP approximation induces a Bayesian linear model in the kernel feature space $\kappa(\cdot)$, with structured weight sharing across the $r$ projected dimensions.

This establishes an explicit equivalence between sparse GP approximations and a class of VBNs, where uncertainty is encoded in a low-dimensional latent weight space while nonlinear expressiveness is provided by kernel-induced features.

### A.4. Proof of Theorem 4.1: exact GP recasting through linear transformations

*Theorem*: The following statements are equivalent:

1) There exists a zero-mean GP $\tilde{g}(\cdot)$ on $\mathbb{R}^r$ such that $g(\boldsymbol{x}) \overset{\mathrm{d}}{=} \tilde{g}(\boldsymbol{A}\boldsymbol{x})$.

2) There exists a covariance function $\tilde{k}(\cdot, \cdot') : \mathbb{R}^r \times \mathbb{R}^r \to \mathbb{R}$ such that $k(\boldsymbol{x}, \boldsymbol{x}') = \tilde{k}(\boldsymbol{A}\boldsymbol{x}, \boldsymbol{A}\boldsymbol{x}')$.

Moreover, when these conditions hold, $\tilde{k}$ is positive semi-definite on the column space $\mathrm{range}(\boldsymbol{A}) \equiv \{\boldsymbol{A}\boldsymbol{x} : \boldsymbol{x} \in \mathbb{R}^n\} \subseteq \mathbb{R}^r$ and therefore defines a valid GP.

*Proof.* We prove the equivalence of (1) and (2) by matching finite-dimensional distributions, and then show the stated positive semi-definiteness claim on $\mathrm{range}(A)$.

**(1) $\Rightarrow$ (2).** Assume there exists a zero-mean GP $\tilde{g}$ on $\mathbb{R}^r$ such that

$$g(\boldsymbol{x}) \stackrel{\mathrm{d}}{=} \tilde{g}(\boldsymbol{A}\boldsymbol{x}) \qquad \text{(as stochastic processes indexed by } \boldsymbol{x} \in \mathbb{R}^n\text{)}.$$

By definition of equality in distribution for stochastic processes, for any $m \in \mathbb{N}$ and any inputs $\boldsymbol{x}_1, \ldots, \boldsymbol{x}_m \in \mathbb{R}^n$, the random vectors

$$\big(g(\boldsymbol{x}_1), \ldots, g(\boldsymbol{x}_m)\big) \quad \text{and} \quad \big(\tilde{g}(\boldsymbol{A}\boldsymbol{x}_1), \ldots, \tilde{g}(\boldsymbol{A}\boldsymbol{x}_m)\big)$$

have the same distribution. Both are centered Gaussian vectors. Hence, their covariance matrices coincide entrywise:

$$\mathrm{Cov}\big(g(\boldsymbol{x}_i), g(\boldsymbol{x}_j)\big) = \mathrm{Cov}\big(\tilde{g}(\boldsymbol{A}\boldsymbol{x}_i), \tilde{g}(\boldsymbol{A}\boldsymbol{x}_j)\big), \qquad \forall i, j \in [m].$$

Let $\tilde{k} : \mathbb{R}^r \times \mathbb{R}^r \to \mathbb{R}$ denote the covariance function of $\tilde{g}$: $\tilde{k}(\boldsymbol{u}, \boldsymbol{v}) = \mathrm{Cov}(\tilde{g}(\boldsymbol{u}), \tilde{g}(\boldsymbol{v}))$. Then for all $\boldsymbol{x}, \boldsymbol{x}' \in \mathbb{R}^n$,

$$k(\boldsymbol{x}, \boldsymbol{x}') := \mathrm{Cov}\big(g(\boldsymbol{x}), g(\boldsymbol{x}')\big) = \mathrm{Cov}\big(\tilde{g}(A\boldsymbol{x}), \tilde{g}(A\boldsymbol{x}')\big) = \tilde{k}(A\boldsymbol{x}, A\boldsymbol{x}').$$

This is exactly statement (2).

**(2) $\Rightarrow$ (1).** Assume there exists a function $\tilde{k} : \mathbb{R}^r \times \mathbb{R}^r \to \mathbb{R}$ such that

$$k(\boldsymbol{x}, \boldsymbol{x}') = \tilde{k}(\boldsymbol{A}\boldsymbol{x}, \boldsymbol{A}\boldsymbol{x}') \qquad \forall \, \boldsymbol{x}, \boldsymbol{x}' \in \mathbb{R}^n. \tag{41}$$

We first show that $\tilde{k}$ is positive semi-definite on the index set $\mathrm{range}(A)$, and hence defines a valid GP on that set.

Take any $m \in \mathbb{N}$, any points $\boldsymbol{u}_1, \ldots, \boldsymbol{u}_m \in \mathrm{range}(\boldsymbol{A})$, and any coefficients $c_1, \ldots, c_m \in \mathbb{R}$. By definition of $\mathrm{range}(\boldsymbol{A})$, for each $i$ there exists $\boldsymbol{x}_i \in \mathbb{R}^n$ such that $\boldsymbol{u}_i = \boldsymbol{A}\boldsymbol{x}_i$. Then, using (41),

$$\sum_{i=1}^{m} \sum_{j=1}^{m} c_i c_j \, \tilde{k}(\boldsymbol{u}_i, \boldsymbol{u}_j) = \sum_{i=1}^{m} \sum_{j=1}^{m} c_i c_j \, \tilde{k}(A\boldsymbol{x}_i, A\boldsymbol{x}_j) = \sum_{i=1}^{m} \sum_{j=1}^{m} c_i c_j \, k(\boldsymbol{x}_i, \boldsymbol{x}_j).$$

Since $k$ is a covariance function of the GP $g$, it is positive semi-definite on $\mathbb{R}^n$, so the right-hand side is $\geq 0$. Therefore,

$$\sum_{i,j} c_i c_j \, \tilde{k}(\boldsymbol{u}_i, \boldsymbol{u}_j) \;\geq\; 0 \qquad \text{for all choices of } \{\boldsymbol{u}_i\} \subset \mathrm{range}(\boldsymbol{A}), \; \{c_i\} \subset \mathbb{R}.$$

Thus $\tilde{k}$ is positive semi-definite on $\mathrm{range}(\boldsymbol{A})$.

By the standard existence theorem for Gaussian processes (e.g., via Kolmogorov extension), there exists a centered GP

$$\tilde{g} : \mathrm{range}(\boldsymbol{A}) \to \mathbb{R}$$

with covariance function $\tilde{k}$ restricted to $\mathrm{range}(\boldsymbol{A}) \times \mathrm{range}(\boldsymbol{A})$, i.e. $\mathrm{Cov}(\tilde{g}(\boldsymbol{u}), \tilde{g}(\boldsymbol{v})) = \tilde{k}(\boldsymbol{u}, \boldsymbol{v})$ for all $\boldsymbol{u}, \boldsymbol{v} \in \mathrm{range}(\boldsymbol{A})$.

Now define a process indexed by $\boldsymbol{x} \in \mathbb{R}^n$ via composition:

$$h(\boldsymbol{x}) := \tilde{g}(\boldsymbol{A}\boldsymbol{x}), \qquad \boldsymbol{x} \in \mathbb{R}^n.$$

For any finite set $\boldsymbol{x}_1, \ldots, \boldsymbol{x}_m \in \mathbb{R}^n$, the vector $(h(\boldsymbol{x}_1), \ldots, h(\boldsymbol{x}_m))$ is centered Gaussian with covariance

$$\mathrm{Cov}\big(h(\boldsymbol{x}_i), h(\boldsymbol{x}_j)\big) = \mathrm{Cov}\big(\tilde{g}(\boldsymbol{A}\boldsymbol{x}_i), \tilde{g}(\boldsymbol{A}\boldsymbol{x}_j)\big) = \tilde{k}(\boldsymbol{A}\boldsymbol{x}_i, \boldsymbol{A}\boldsymbol{x}_j) = k(\boldsymbol{x}_i, \boldsymbol{x}_j),$$

where the last equality is (41). Hence the finite-dimensional distributions of $h$ match those of $g$ (both centered Gaussian with identical covariance matrices), implying

$$g(\boldsymbol{x}) \stackrel{\mathrm{d}}{=} h(\boldsymbol{x}) = \tilde{g}(A\boldsymbol{x}) \quad \text{as stochastic processes on } \mathbb{R}^n.$$

This is statement (1).

The argument above already shows the claimed property: whenever (41) holds, $\tilde{k}$ is positive semi-definite on $\mathrm{range}(\boldsymbol{A}) = \{\boldsymbol{A}\boldsymbol{x} : \boldsymbol{x} \in \mathbb{R}^n\} \subseteq \mathbb{R}^r$ and therefore defines a valid GP on that index set. $\qquad \square$

## A.5. Amortized variational inference: alleviating the overconfidence issue

In this section, we explain why amortized variational inference can alleviate the overconfidence issue commonly observed in variational Bayesian models, including sparse variational GPs and Bayesian neural networks, from the perspective of the evidence lower bound (ELBO).

Overconfidence as a serious concern in Bayesian learning, especially in DKL, has been rigorously investigated in previous work. Consider a latent variable model with latent variables $z$ and observations $\mathcal{D} = \{(x_n, y_n)\}_{n=1}^N$. Variational inference seeks to approximate the true posterior $p(z \mid \mathcal{D})$ by a variational distribution $q(z)$ via the ELBO:

$$\mathcal{L}_{\text{ELBO}} = \mathbb{E}_{q(z)}[\log p(\mathcal{D} \mid z)] - \text{KL}\big(q(z) \,\|\, p(z)\big). \tag{42}$$

**Overconfidence in standard variational inference**: In standard (non-amortized) variational inference, the variational parameters of $q(z)$ are optimized independently for a given dataset. When expressive likelihood models are used (e.g., deep networks or flexible kernels), the ELBO objective admits solutions where:

- the likelihood term strongly favors sharp posterior modes that fit the data closely;

- the KL regularizer is insufficient to prevent posterior collapse toward overly concentrated distributions, especially under mean-field assumptions.

As a result, the learned posterior $q(z)$ may significantly underestimate posterior uncertainty, leading to overconfident predictions, particularly under distribution shift or in regions with limited data support. From an ELBO perspective, this phenomenon arises because the optimization is performed *pointwise* in the variational parameters: the posterior is allowed to specialize aggressively to the training data without being constrained to generalize across inputs.

Amortized variational inference introduces a shared inference model $q_\phi(z \mid x)$, parameterized by $\phi$, typically implemented as a neural network. The ELBO becomes

$$\mathcal{L}_{\text{A-ELBO}} = \sum_{n=1}^N \mathbb{E}_{q_\phi(z \mid x_n)}[\log p(y_n \mid x_n, z)] - \text{KL}\big(q_\phi(z \mid x_n) \,\|\, p(z)\big). \tag{43}$$

**Why amortized variational inference mitigates overconfidence**: Crucially, amortization ties the variational parameters across all data points through shared mapping $\phi$. This coupling introduces an implicit regularization effect:

- The variational posterior can no longer overfit individual data points independently;

- Sharp posterior concentrations must be explainable by a smooth inference function $q_\phi(z \mid x)$ that generalizes across inputs;

- Excessively confident posteriors that are inconsistent across nearby inputs are penalized indirectly, as they increase the overall ELBO loss.

This regularizing effect is particularly beneficial for uncertainty quantification. Under amortized inference, epistemic uncertainty is shaped not only by the local likelihood fit, but also by how well uncertainty patterns can be represented by the shared inference network. As a result, uncertainty is naturally preserved in out-of-distribution or data-sparse regions, where confident predictions would require extrapolations unsupported by the amortized posterior.

In the context of sparse variational GPs or their Bayesian network interpretations (Appendix A.3), amortized inference over low-dimensional latent variables (e.g., inducing weights or kernel coefficients) further strengthens this effect by limiting the capacity of the posterior to overfit. This provides a theoretically guided explanation for why amortized variational inference often yields better-calibrated uncertainty compared to non-amortized or pointwise variational approaches.

# B. Experimental Details

## B.1. Intuitive example: LoRA regression

**Goal.** We construct a 1D regression toy problem to visualize posterior mean and uncertainty after fine-tuning with Bayesian LoRA variants. The key diagnostic is the empirical coverage of nominal confidence intervals under an input-domain shift:

overconfident methods yield narrow intervals with low coverage, while well-calibrated methods maintain appropriate interval width in regions with limited evidence.

**Data generation.** Inputs are one-dimensional, $x \in \mathbb{R}$. We sample a latent function $f(\cdot)$ from a Gaussian process,

$$f \sim \mathcal{GP}(0, k_{\text{RBF}}(x, x')), \qquad k_{\text{RBF}}(x, x') = \sigma_f^2 \exp\left(-\frac{(x - x')^2}{2\ell^2}\right), \tag{44}$$

and generate observations with homoscedastic noise

$$y = f(x) + \epsilon, \qquad \epsilon \sim \mathcal{N}(0, \sigma_n^2). \tag{45}$$

Unless stated otherwise, we use $\ell = 1.0$, $\sigma_f = 1.0$, and $\sigma_n = 0.2$. We fix one GP draw $f$ for the entire experiment (shared across all methods) to ensure comparability.

**Pretraining vs fine-tuning splits.** We create two domains to mimic a distribution shift.

*Source (pretraining) domain:* $x \sim \text{Unif}[-3, 3]$. We draw $N_{\text{src}} = 200$ training points and $N_{\text{src}}^{\text{test}} = 400$ test points from the same range. These correspond to the black (train) and red (test) points in Fig. 5(a).

*Target (fine-tuning) domain:* $x \sim \text{Unif}[-5, 5]$ (extrapolation / covariate shift). We draw $N_{\text{tgt}} = 200$ fine-tuning training points and $N_{\text{tgt}}^{\text{test}} = 400$ fine-tuning test points from this shifted support, corresponding to the blue (FT train) and green (FT test) points in Fig. 5(a).

All inputs are standardized using the source training statistics: $\tilde{x} = (x - \mu_{\text{src}})/s_{\text{src}}$. Targets are kept in the original scale for visualization.

**Backbone (pretrained) model.** The pretrained predictor is a two-layer MLP with hidden width $d = 64$ and activation $\phi(\cdot) = \text{ReLU}$. The backbone is pretrained on the source domain by minimizing MSE with Adam (learning rate $10^{-3}$, batch size 64, 5,000 steps). After pretraining, *all backbone weights are frozen* and only LoRA (or Bayesian LoRA) parameters are updated during fine-tuning.

**LoRA parameterization.** We attach LoRA to all linear layers (including the output head) for clarity in the 1D visualization. Let the frozen weight be $W_i \in \mathbb{R}^{1 \times d}$. LoRA introduces a low-rank update

$$W_i' = W_i + \Delta W_i, \qquad \Delta W_i = B_i A_i, \quad i \in \{1, 2\}, \tag{46}$$

where $A \in \mathbb{R}^{r \times d_i}$ and $B \in \mathbb{R}^{d_{i+1} \times r}$, and rank $r = 2$ in all toy experiments. During fine-tuning, we optimize only the LoRA parameters; the backbone remains fixed.

**BLoB:** a Bayesian LoRA baseline where the adapted parameters follow a mean-field variational posterior (diagonal Gaussian) and are trained by maximizing an ELBO on the target training set. Predictive uncertainty is obtained by Monte Carlo sampling of LoRA parameters at test time (e.g., $T = 50$ samples).

**C-LoRA:** a contextual Bayesian LoRA variant where the variational parameters are produced by a lightweight amortization network conditioned on the layer input (or a summary of it), still under a mean-field posterior family. In the toy setup, the context network is a single hidden-layer MLP with $\text{tanh}$ activation.

**GPan-LoRA (ours):** we model the Bayesian uncertainty in the low-rank adapted space via a GP-style correlated posterior over the rank-$r$ coordinates, trained with amortized variational inference. Concretely, we parameterize the variational distribution with structured covariance in the $r$-dimensional subspace (rather than a fully factorized posterior in the ambient dimension), enabling calibrated uncertainty while retaining scalability. We use 32 inducing variables in the $r$-space and optimize the variational objective on the target training set.

**Fine-tuning protocol (all Bayesian methods).** All Bayesian LoRA methods are fine-tuned on the target training set for 2,000 steps using Adam (learning rate $5 \times 10^{-4}$, batch size 64). We apply KL warm-up on the ELBO term: the KL weight $\beta$ increases linearly from 0 to 1 during the first 30% of updates, and remains 1 afterwards. For MC-based predictive estimates, we use $T = 50$ posterior samples for BLoB and C-LoRA; GPan-LoRA uses its closed-form / GP-based predictive where applicable, otherwise also uses $T = 1000$ samples for fair plotting.

**GP ground-truth posterior (oracle).** To provide the reference uncertainty in Fig. 5(c), we compute the exact GP posterior under the same kernel used to generate $f$ (RBF with known hyperparameters). Given training data $(X, y)$, the posterior

predictive at $x_*$ is

$$p(y_* \mid x_*, X, y) = \mathcal{N}\big(\mu_*(x_*), \sigma_*^2(x_*)\big), \tag{47}$$

with standard closed-form expressions. The plotted $95\%$ CI is $\mu_* \pm 1.96\,\sigma_*$.

**Evaluation metrics.** We report RMSE and NLL on both the source test set and the target test set, and emphasize calibration via empirical coverage of the nominal $95\%$ CI:

$$\mathrm{Cov}_{95} = \frac{1}{N} \sum_{i=1}^{N} \mathbb{I}[y_i \in [\mu(x_i) - 1.96\sigma(x_i),\ \mu(x_i) + 1.96\sigma(x_i)]]. \tag{48}$$

Overconfidence is reflected by $\mathrm{Cov}_{95} \ll 0.95$ (intervals too narrow), which is the behavior observed for BLoB and C-LoRA in Fig. 5(d,e), while GPan-LoRA maintains coverage closer to the nominal level (Fig. 5(f)).

**Plotting details.** All curves are evaluated on a dense grid of $x$ over $[-5, 5]$ (e.g., 400 evenly spaced points). We visualize the posterior mean (red line) and the $95\%$ predictive interval (shaded region), and overlay training points as black dots for the corresponding domain in each panel. We use the same random seed, dataset realization, and pretrained backbone across all compared methods.

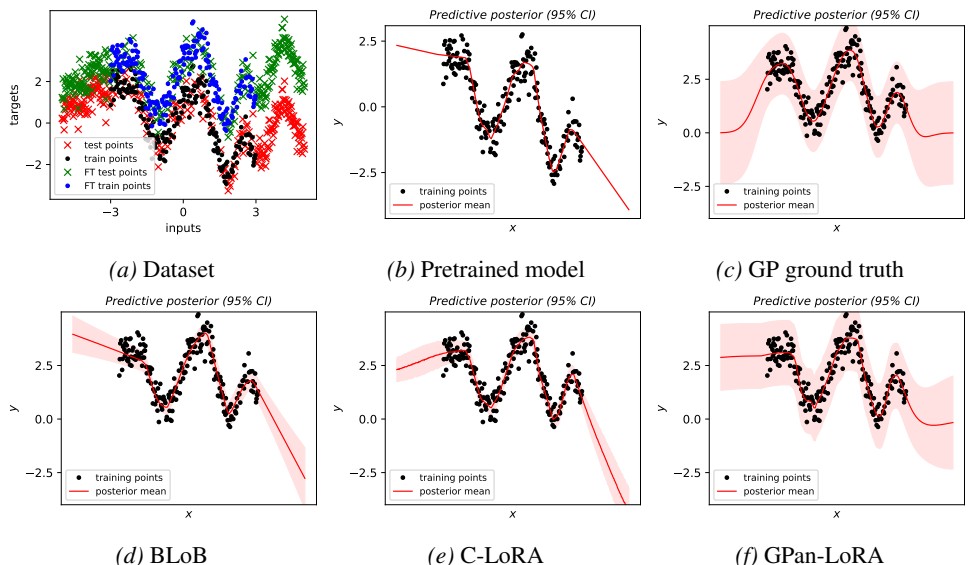

*(a)* Dataset      *(b)* Pretrained model      *(c)* GP ground truth

*(d)* BLoB      *(e)* C-LoRA      *(f)* GPan-LoRA

*Figure 5.* Fine-tuning a regression dataset with Bayesian LoRA. BLoB and C-LoRA exhibit low empirical coverage of confidence intervals, indicating overconfidence, while GPan-LoRA produces well-calibrated uncertainty estimates.

*Table 4.* MSE and NLL for fine-tuning regression datasets (ID and OOD) with Bayesian LoRA. Lower is better.

| Method | In-distribution | | Out-of-distribution | |
|---|---|---|---|---|
| | MSE $\downarrow$ | NLL $\downarrow$ | MSE $\downarrow$ | NLL $\downarrow$ |
| BLoB | $0.21 \pm 0.00$ | $0.64 \pm 0.11$ | $2.48 \pm 0.07$ | $10.34 \pm 1.82$ |
| C-LoRA | $0.19 \pm 0.00$ | $0.78 \pm 0.01$ | $6.32 \pm 0.05$ | $6.77 \pm 0.06$ |
| GPan-LoRA | $0.19 \pm 0.00$ | $-0.11 \pm 0.00$ | $3.50 \pm 0.00$ | $1.98 \pm 0.00$ |

## B.2. Implementation Details

**Dataset.** We summarize the downstream datasets and the corresponding prompt templates used for fine-tuning in Table 6. All tasks are formulated as multiple-choice or binary classification problems using unified, instruction-style prompts to ensure a consistent training and evaluation protocol across datasets.

*Table 5.* Hyperparameter settings for `LLaMA3.1-8B` and `Qwen3-8B` experiments.

| Hyperparameter | LLaMA-3.1-8B | Qwen-3-8B |
|---|---|---|
| Max Seq. Length | 300 | 512 |
| LoRA Rank ($r$) | 8 | 8 |
| LoRA Alpha ($\alpha$) | 16 | 16 |
| Target Modules | $W_q, W_v$ $W_{lm}$ | $W_q, W_k, W_v$ $W_{gate}, W_{up}, W_{down}$ |
| Training Iterations | | 5,000 |
| Batch Size | | 4 |
| Eval Interval | | 500 |

Table 7 reports the size of the training split and the label space for each dataset. The considered benchmarks span a diverse range of commonsense reasoning settings, including binary (e.g., BoolQ, Winogrande) and multi-class classification (e.g., ARC, OBQA, MMLU), enabling a comprehensive evaluation of model performance and uncertainty behavior under varying label cardinalities and data scales.

*Table 6.* Prompts for common-sense reasoning tasks used for fine-tuning

| Downstream task | Prompt template |
|---|---|
| Winogrande (WG-S/WG-M) | Select one of the choices that answers the following question: {question} Choices: A. {option1}. B. {option2}. Answer: |
| ARC (ARC-C/ARC-E), Openbook QA (OBQA), MMLU | Select one of the choices that answers the following question: {question} Choices: A. {choice1}. B. {choice2}, C. {choice3}. D. {choice4}. Answer: |
| BoolQ | Answer the question with only True or False: {question} Context: {passage} |

*Table 7.* Training set size and label space size for each dataset

| | WG-S | ARC-C | ARC-E | WG-M | OBQA | BoolQ |
|---|---|---|---|---|---|---|
| Number of training samples | 512 | 892 | 1.8K | 2.05k | 3.97k | 1.99k |
| Label space size | 2 | 5 | 5 | 2 | 4 | 2 |

**Uncertainty Quantification Evaluation Metrics.** The quality of uncertainty quantification is commonly assessed using negative log-likelihood (NLL) and expected calibration error (ECE), which respectively capture model overconfidence and the alignment between predictive confidence and accuracy. For a model $P_\theta$, the NLL over a dataset of size $N$ is defined as

$$\text{NLL} = \frac{1}{N} \sum_{i=1}^{N} -\log P_\theta(y_i \mid x_i), \tag{49}$$

which penalizes low predictive probability assigned to the correct label $y_i$. In particular, overconfident but incorrect predictions incur a large penalty, leading to an increased NLL. In addition to NLL, ECE measures model calibration by quantifying the discrepancy between confidence and accuracy. Predictions are first grouped into $M$ bins $\{B_m\}_{m=1}^{M}$ based on their confidence levels, and ECE is computed as

$$\text{ECE} = \sum_{m=1}^{M} \frac{|B_m|}{N} |\text{acc}(B_m) - \text{conf}(B_m)|, \tag{50}$$

where $\text{acc}(B_m)$ and $\text{conf}(B_m)$ denote the average accuracy and confidence within bin $B_m$, respectively:

$$\text{acc}(B_m) = \frac{1}{|B_m|} \sum_{i \in B_m} \mathbf{1}(\hat{y}_i = y_i), \quad \text{conf}(B_m) = \frac{1}{|B_m|} \sum_{i \in B_m} P_\theta(\hat{y}_i \mid x_i). \tag{51}$$

**Base Models and Sequence Length.** We conduct experiments using two base models `LLaMA3.1-8B` and `Qwen3-8B`. To account for architectural and tokenizer differences between the two models, we adopt model-specific maximum sequence lengths, setting the maximum sequence length to 300 for `LLaMA3.1-8B` and 512 for `Qwen3-8B`.

**LoRA Injection.**   We apply LoRA adapters to `LLaMA3.1-8B` and `Qwen3-8B` with model-specific injection configurations. For `LLaMA3.1-8B`, LoRA adapters are applied to the query and value projection layers in the self-attention module, as well as to the final language modeling head that maps hidden states to vocabulary logits. For `Qwen3-8B`, LoRA adapters are applied to all linear projection layers within each Transformer block. This includes the query, key, value, and output projections in the self-attention module, as well as the gate, up, and down projections in the feed-forward network.

**Baseline Implementations.**   We compare GPan-LoRA with several deterministic and Bayesian LoRA baselines, including MLE, Monte Carlo Dropout (MCD), Deep Ensembles, Laplace-LoRA (LAP), BLoB, and C-LoRA. Except for LAP, all baselines are implemented using the unified Bayesian-PEFT framework[2], ensuring consistent model architectures, data preprocessing, optimization settings, and evaluation protocols across methods. Note that for BLoB, C-LoRA, and GPan-LoRA, we draw one posterior sample per forward pass during training, and report results by averaging predictions over 10 posterior samples at evaluation. Deep Ensembles are implemented following prior work, using an ensemble size of $N = 3$ independently trained LoRA models with different random seeds, whose predictions are aggregated at inference time. Monte Carlo Dropout (MCD) is implemented by enabling dropout layers during inference and averaging predictions over multiple stochastic forward passes. Laplace-LoRA is implemented using the official Laplace-LoRA repository[3], employing a Laplace approximation with a Kronecker-factored log-likelihood Hessian to approximate the posterior precision. The approximation is applied to all trainable parameters, following the default settings of the official implementation. As Laplace-LoRA is applied as a post-hoc Bayesian approximation on a pretrained deterministic LoRA model, its number of trainable parameters matches that of deterministic LoRA. C-LoRA is implemented using its official repository[4], following the authors' recommended configurations. For BLoB, we apply gradient clipping to stabilize training and prevent posterior collapse on small-scale datasets with limited samples. All remaining baseline configurations strictly follow their respective official implementations.

**Training Protocol.**   All models are trained for 5,000 iterations with a batch size of 4. We evaluate model performance on a held-out validation set every 500 iterations. To reduce potential bias from overfitting—which is particularly important for uncertainty quantification—we select the checkpoint with the highest validation accuracy. Each experiment is repeated three times with different random seeds, and we report the mean and standard deviation of the results.

**Bayesian LoRA Variants.**   For variational inference based Bayesian LoRA methods, including C-LoRA, BLoB, and our proposed **GPan-LoRA**, the coefficient of the KL divergence term significantly affects training stability. We tune this coefficient on the validation set over the candidate set $\{1, 5e^{-1}, 1e^{-1}, 5e^{-2}, 1e^{-2}, 5e^{-2}, 1e^{-3}\}$. We report results using the coefficient that achieves the best ECE on the validation set. Table 5 summarizes the specific hyperparameters used in our experiments.

### B.3. Visualization Details

**Visualization Protocol.** Similar to our previous out-of-distribution (OOD) evaluation, we select 20 samples each from one in-distribution dataset, *OBQA*, and four OOD datasets: *ARC-E* and *ARC-C*, and *Chem* and *Phy*. For each input sample, we perform 50 stochastic forward passes to sample the latent embeddings from the Bayesian posterior. We then apply PCA (Jolliffe & Cadima, 2016) and t-SNE (Maaten & Hinton, 2008) for dimensionality reduction sequentially. The uncertainty of each sample is represented by the variance of its sampled embeddings, visualized as shaded covariance ellipses in Figure 4.

## C. Additional Experiments

### C.1. Computational Efficiency Analysis

Table 8 reports the model size and peak GPU memory consumption of the proposed methods and baselines during training on the WG-S dataset. All experiments were conducted on a single NVIDIA H200 GPU. Laplace approximation is applied *post hoc* to a trained deterministic LoRA model; since it does not introduce additional trainable parameters, we report the same number of trainable parameters as deterministic LoRA (MLE). Although LAP is a competitive baseline, its computational

---

[2] https://github.com/Wang-ML-Lab/bayesian-peft
[3] https://github.com/adamxyang/laplace-lora
[4] https://github.com/ahra99/c_lora

*Table 8.* Computational efficiency comparison. Values in parentheses denote the relative overhead w.r.t. MLE.

| Method | Trainable Params (M) | Peak Memory (MB) | Inference Time (s) |
|---|---|---|---|
| MLE | 4.47 $(\times 1.00)$ | 11,324 $(\times 1.00)$ | 0.093 $(\times 1.00)$ |
| Deep Ensemble | 13.40 $(\times 3.00)$ | 11,324 $(\times 1.00)$ | 2.812 $(\times 30.24)$ |
| Laplace (LAP) | 4.47 $(\times 1.00)$ | 33,121 $(\times 2.93)$ | 0.528 $(\times 5.68)$ |
| BLoB | 6.60 $(\times 1.48)$ | 11,868 $(\times 1.05)$ | 1.193 $(\times 12.83)$ |
| C-LoRA | 5.04 $(\times 1.13)$ | 11,280 $(\times 1.00)$ | 1.210 $(\times 13.01)$ |
| GPan-LoRA | 8.33 $(\times 1.86)$ | 11,363 $(\times 1.00)$ | 1.901 $(\times 20.44)$ |
| LL GPan-LoRA | 4.53 $(\times 1.01)$ | 11,337 $(\times 1.00)$ | 0.935 $(\times 10.05)$ |

*Table 9.* **Kernel sensitivity analysis of GPan-LoRA on** `LLaMA3.1-8B` **pre-trained weights.** We compare the default squared exponential (SE) kernel with a Matérn kernel under the same experimental protocol. Accuracy (**ACC**) and Expected Calibration Error (**ECE**) are reported in percentages, where "↑" and "↓" indicate that higher and lower values are preferred, respectively. The left six columns correspond to the in-distribution setting, while the right four columns report performance under distribution shift. The results indicate that GPan-LoRA is relatively robust to the kernel choice.

| Metric | Method | In-Distribution Datasets | | | | | | Out-of-Distribution Datasets (OBQA→X) | | | |
| | | | | | | | | Small Dist. Shift | | Large Dist. Shift | |
| | | WG-S | ARC-C | ARC-E | WG-M | OBQA | BoolQ | ARC-C | ARC-E | Chem | Phy |
|---|---|---|---|---|---|---|---|---|---|---|---|
| ACC (↑) | GPan-LoRA (SE) | $72.41_{\pm 0.24}$ | $80.06_{\pm 1.01}$ | $91.84_{\pm 0.61}$ | $79.82_{\pm 0.44}$ | $86.66_{\pm 0.94}$ | $87.20_{\pm 3.29}$ | $75.78_{\pm 1.36}$ | $82.74_{\pm 1.23}$ | $48.00_{\pm 2.00}$ | $42.33_{\pm 0.57}$ |
| | GPan-LoRA (Matérn) | $74.13_{\pm 2.61}$ | $81.19_{\pm 1.86}$ | $91.96_{\pm 0.66}$ | $78.08_{\pm 0.70}$ | $85.53_{\pm 0.75}$ | $88.63_{\pm 0.49}$ | $76.23_{\pm 2.44}$ | $83.21_{\pm 0.56}$ | $48.66_{\pm 4.50}$ | $39.33_{\pm 2.08}$ |
| ECE (↓) | GPan-LoRA (SE) | $13.39_{\pm 2.67}$ | $4.60_{\pm 0.88}$ | $2.36_{\pm 0.85}$ | $6.85_{\pm 1.44}$ | $3.12_{\pm 0.64}$ | $1.45_{\pm 0.62}$ | $6.46_{\pm 0.87}$ | $3.31_{\pm 0.37}$ | $11.31_{\pm 2.75}$ | $13.29_{\pm 1.20}$ |
| | GPan-LoRA (Matérn) | $12.82_{\pm 3.20}$ | $5.48_{\pm 1.70}$ | $2.76_{\pm 0.82}$ | $5.00_{\pm 0.61}$ | $2.77_{\pm 0.17}$ | $1.38_{\pm 0.38}$ | $6.18_{\pm 2.19}$ | $3.62_{\pm 0.43}$ | $12.58_{\pm 2.40}$ | $17.76_{\pm 2.49}$ |
| NLL (↓) | GPan-LoRA (SE) | $0.69_{\pm 0.08}$ | $0.51_{\pm 0.02}$ | $0.24_{\pm 0.01}$ | $0.48_{\pm 0.01}$ | $0.37_{\pm 0.02}$ | $0.30_{\pm 0.06}$ | $0.60_{\pm 0.03}$ | $0.46_{\pm 0.01}$ | $1.19_{\pm 0.01}$ | $1.25_{\pm 0.01}$ |
| | GPan-LoRA (Matérn) | $0.72_{\pm 0.11}$ | $0.49_{\pm 0.02}$ | $0.23_{\pm 0.01}$ | $0.48_{\pm 0.01}$ | $0.38_{\pm 0.04}$ | $0.27_{\pm 0.01}$ | $0.58_{\pm 0.01}$ | $0.45_{\pm 0.02}$ | $1.19_{\pm 0.05}$ | $1.25_{\pm 0.01}$ |

overhead is substantially higher than that of other methods, including GPan-LoRA. Among variational inference-based approaches, C-LoRA and GPan-LoRA exhibit the smallest model sizes. This result highlights that GPan-LoRA achieves competitive memory efficiency while retaining a lightweight parameterization.

### C.2. Kernel Sensitivity Analysis

Our main experiments use the squared exponential (SE) kernel as the default kernel choice in GPan-LoRA. However, the proposed framework is not restricted to a specific kernel form: the low-rank GP formulation can incorporate any positive semi-definite kernel that is well-defined in the projected LoRA feature space. To examine whether the empirical performance is sensitive to this choice, we conduct an additional kernel ablation by replacing the SE kernel with a Matérn kernel while keeping the remaining training and evaluation settings unchanged.

Table 9 reports the results on both in-distribution datasets and distribution-shifted OOD datasets. Overall, the Matérn kernel yields performance comparable to the SE kernel across ACC, ECE, and NLL. In several in-distribution settings, Matérn slightly improves ACC or calibration, while SE remains stronger on some large-shift OOD calibration metrics such as Phy ECE. Importantly, the differences are generally modest, suggesting that the uncertainty benefits of GPan-LoRA mainly come from the low-rank GP Bayesian formulation and amortized variational inference rather than from a carefully tuned kernel choice. This supports the kernel-agnostic nature of our framework and indicates that GPan-LoRA is relatively robust to reasonable kernel choices.

### C.3. Seed-Level Paired Significance Tests

To strengthen the empirical evidence beyond mean±std summaries, we further conduct seed-level paired $t$-tests using the raw per-seed results reported in Table 2. Since the primary goal of GPan-LoRA is uncertainty quantification rather than maximizing accuracy, we focus this statistical analysis on calibration-oriented metrics, namely ECE and NLL. Accuracy results are reported in Table 2 to demonstrate competitive task performance.

For each baseline, metric, dataset, and random seed, we compute the paired improvement between GPan-LoRA and the baseline under the same evaluation setting. For ECE and NLL, where lower values are preferred, we define

$$\Delta = \text{Metric}_{\text{baseline}} - \text{Metric}_{\text{GPan}}. \tag{52}$$

*Table 10.* Seed-level paired $t$-test statistics comparing GPan-LoRA with baselines on uncertainty-oriented metrics under distribution shift. The 'OOD' group follows the settings adopted in Table 2 and includes OBQA→{ARC-C, ARC-E, Chem, Phy}; the 'Large Dist. Shift' group includes OBQA→{Chem, Phy}. Each paired observation corresponds to one dataset–seed pair. For ECE and NLL, improvement is defined as $\Delta = \text{Metric}_{\text{baseline}} - \text{Metric}_{\text{GPan}}$, where positive $\Delta$ indicates better performance of GPan-LoRA. 'Win/n' denotes the number of dataset–seed pairs where GPan-LoRA improves over the baseline. We report the mean paired improvement, 95% confidence interval, and Holm-corrected $p$-value.

| Metric | Comparison | Out-of-Distribution Datasets | | | | | Large Dist. Shift Datasets | | | | |
|---|---|---|---|---|---|---|---|---|---|---|---|
| | | $n$ | Win/n | Mean $\Delta$ | 95% CI | $p_{\text{Holm}}$ | $n$ | Win/n | Mean $\Delta$ | 95% CI | $p_{\text{Holm}}$ |
| ECE | GPan-LoRA vs. MLE | 12 | 12/12 | +16.82 | [+11.17, +22.48] | < .001 | 6 | 6/6 | +24.80 | [+20.11, +29.50] | < .001 |
| | GPan-LoRA vs. LAP | 12 | 6/12 | +0.67 | [−0.80, +2.14] | .340 | 6 | 5/6 | +1.99 | [−0.31, +4.30] | .077 |
| | GPan-LoRA vs. BLoB | 12 | 12/12 | +8.96 | [+3.85, +14.08] | .005 | 6 | 6/6 | +16.00 | [+11.11, +20.89] | .001 |
| | GPan-LoRA vs. C-LoRA | 12 | 12/12 | +6.52 | [+3.31, +9.73] | .003 | 6 | 6/6 | +10.23 | [+5.43, +15.04] | .006 |
| NLL | GPan-LoRA vs. MLE | 12 | 12/12 | +0.68 | [+0.47, +0.89] | < .001 | 6 | 6/6 | +0.95 | [+0.70, +1.20] | < .001 |
| | GPan-LoRA vs. LAP | 12 | 6/12 | −0.02 | [−0.08, +0.04] | .452 | 6 | 6/6 | +0.06 | [+0.02, +0.10] | .012 |
| | GPan-LoRA vs. BLoB | 12 | 10/12 | +0.29 | [+0.10, +0.47] | .017 | 6 | 6/6 | +0.54 | [+0.38, +0.70] | .001 |
| | GPan-LoRA vs. C-LoRA | 12 | 9/12 | +0.10 | [+0.02, +0.18] | .040 | 6 | 6/6 | +0.21 | [+0.13, +0.29] | .002 |

Thus, positive $\Delta$ consistently indicates better performance of GPan-LoRA. This paired design compares methods under the same dataset and random seed, reducing variability due to different data splits or random initialization.

Because each individual dataset has only three random seeds, we avoid drawing per-dataset significance claims. Instead, we aggregate paired differences across dataset–seed pairs within each evaluation split. The Out-of-Distribution (OOD) analysis uses 12 paired observations from four OOD datasets and three random seeds, while the Large OOD analysis uses 6 paired observations from Chem and Phy across three random seeds. We apply two-sided paired $t$-tests and use Holm correction to account for multiple baseline comparisons within each metric and split.

As shown in Table 10, GPan-LoRA provides consistent improvements in uncertainty-oriented metrics under distribution shift. Compared with MLE, BLoB, and C-LoRA, GPan-LoRA significantly reduces both ECE and NLL on OOD datasets, with particularly strong improvements under large distribution shifts. Compared with LAP, the overall OOD differences are smaller and not uniformly significant, but GPan-LoRA achieves a consistent NLL improvement under large OOD shifts. These results support our main claim that GPan-LoRA improves calibration and probabilistic prediction quality under distribution shift, while avoiding over-claiming significance on raw accuracy.

## C.4. Evaluation on Qwen3-8B

To examine whether the conclusions drawn in the main paper generalize beyond a single base model, we additionally evaluate all uncertainty estimation methods on `Qwen3-8B` pre-trained weights, using the same experimental protocol, datasets, and evaluation metrics as in Table 2.

As shown in Table 11, the overall trends are highly consistent with those observed on `LLaMA3.1-8B`. In particular, GPan-LoRA continues to demonstrate competitive predictive accuracy while achieving substantially improved calibration and negative log-likelihood, especially under distribution shift. These results suggest that the advantages of GPan-LoRA are not specific to a particular backbone, but hold across different large language model architectures.

*Table 11.* **Performance of different uncertainty estimate methods applied to LoRA on** `Qwen3-8B` **pre-trained weights**, where Accuracy (**ACC**) and Expected Calibration Error (**ECE**) are reported in percentages. "↑" and "↓" indicate that higher and lower values are preferred, respectively. The left six columns demonstrate the UQ performance of in-distribution setting, and the right four columns is the UQ performance under distribution shift. Best and second-best ECE and NLL results are highlighted in **boldface** and underline, respectively.

| Metric | Method | In-Distribution Datasets | | | | | | Out-of-Distribution Datasets (OBQA→X) | | | |
| | | | | | | | | *Small Dist. Shift* | | *Large Dist. Shift* | |
| | | WG-S | ARC-C | ARC-E | WG-M | OBQA | BoolQ | ARC-C | ARC-E | Chem | Phy |
| ACC (↑) | MLE | $73.91_{\pm0.73}$ | $88.73_{\pm0.85}$ | $96.59_{\pm0.20}$ | $81.67_{\pm0.29}$ | $89.46_{\pm0.41}$ | $89.85_{\pm0.22}$ | $88.28_{\pm0.39}$ | $94.83_{\pm0.10}$ | $55.66_{\pm3.05}$ | $54.33_{\pm4.04}$ |
| | MCD | $75.23_{\pm0.89}$ | $87.61_{\pm0.78}$ | $96.36_{\pm0.20}$ | $82.46_{\pm0.98}$ | $91.53_{\pm0.23}$ | $90.00_{\pm0.06}$ | $90.37_{\pm0.23}$ | $95.42_{\pm0.00}$ | $53.50_{\pm2.12}$ | $58.00_{\pm2.82}$ |
| | DeepEns | $74.55_{\pm0.78}$ | $88.51_{\pm1.21}$ | $96.71_{\pm0.26}$ | $81.80_{\pm0.49}$ | $90.46_{\pm0.83}$ | $90.08_{\pm0.17}$ | $89.86_{\pm0.67}$ | $96.01_{\pm0.10}$ | $56.66_{\pm1.52}$ | $57.33_{\pm2.10}$ |
| | LAP | $73.27_{\pm0.43}$ | $90.96_{\pm1.57}$ | $96.36_{\pm0.27}$ | $79.79_{\pm0.34}$ | $90.87_{\pm0.42}$ | $89.20_{\pm0.46}$ | $90.85_{\pm0.68}$ | $95.88_{\pm0.44}$ | $56.33_{\pm1.15}$ | $56.54_{\pm2.47}$ |
| | BLoB | $73.52_{\pm0.71}$ | $88.06_{\pm1.08}$ | $96.65_{\pm0.07}$ | $82.08_{\pm0.72}$ | $91.06_{\pm0.30}$ | $89.95_{\pm0.30}$ | $89.18_{\pm0.47}$ | $95.95_{\pm0.74}$ | $53.50_{\pm2.12}$ | $60.50_{\pm0.70}$ |
| | C-LoRA | $75.94_{\pm0.71}$ | $87.95_{\pm0.39}$ | $95.95_{\pm0.24}$ | $79.37_{\pm2.73}$ | $90.60_{\pm0.34}$ | $90.08_{\pm0.11}$ | $90.09_{\pm0.85}$ | $95.59_{\pm0.30}$ | $55.33_{\pm1.52}$ | $59.00_{\pm1.00}$ |
| | GPan-LoRA | $75.29_{\pm1.91}$ | $85.47_{\pm2.86}$ | $95.89_{\pm0.44}$ | $80.77_{\pm0.82}$ | $89.86_{\pm0.23}$ | $89.84_{\pm0.15}$ | $86.03_{\pm0.70}$ | $93.72_{\pm0.96}$ | $57.33_{\pm0.57}$ | $58.00_{\pm2.00}$ |
| ECE (↓) | MLE | $23.12_{\pm1.74}$ | $6.20_{\pm0.99}$ | $2.11_{\pm0.26}$ | $13.37_{\pm0.55}$ | $4.69_{\pm0.55}$ | $4.00_{\pm1.77}$ | $10.46_{\pm0.87}$ | $4.42_{\pm0.23}$ | $27.76_{\pm5.02}$ | $33.64_{\pm3.31}$ |
| | MCD | $18.16_{\pm4.44}$ | $9.48_{\pm0.20}$ | $2.22_{\pm0.66}$ | $14.74_{\pm1.66}$ | $7.00_{\pm1.89}$ | $4.42_{\pm4.52}$ | $8.29_{\pm0.11}$ | $3.73_{\pm0.37}$ | $31.07_{\pm0.11}$ | $29.63_{\pm0.53}$ |
| | DeepEns | $24.12_{\pm1.06}$ | $9.68_{\pm0.92}$ | $2.64_{\pm0.22}$ | $15.45_{\pm1.28}$ | $7.53_{\pm0.35}$ | $3.68_{\pm2.04}$ | $5.92_{\pm0.64}$ | $2.13_{\pm0.12}$ | $19.40_{\pm0.25}$ | $24.28_{\pm3.22}$ |
| | LAP | $\mathbf{4.95_{\pm0.77}}$ | $\mathbf{3.61_{\pm0.83}}$ | $\underline{1.70_{\pm0.84}}$ | $\mathbf{2.90_{\pm1.10}}$ | $\underline{2.59_{\pm0.43}}$ | $\underline{1.67_{\pm0.82}}$ | $\underline{3.09_{\pm0.57}}$ | $2.05_{\pm0.32}$ | $\underline{14.78_{\pm1.14}}$ | $18.95_{\pm2.39}$ |
| | BLoB | $11.57_{\pm0.15}$ | $\underline{5.58_{\pm3.09}}$ | $1.98_{\pm0.59}$ | $11.74_{\pm0.69}$ | $5.55_{\pm0.48}$ | $6.76_{\pm0.31}$ | $6.75_{\pm1.43}$ | $2.37_{\pm0.30}$ | $29.19_{\pm4.00}$ | $27.49_{\pm1.99}$ |
| | C-LoRA | $\underline{10.58_{\pm3.69}}$ | $6.68_{\pm1.16}$ | $2.19_{\pm0.40}$ | $7.84_{\pm2.71}$ | $4.23_{\pm1.61}$ | $2.08_{\pm0.53}$ | $5.06_{\pm2.05}$ | $\mathbf{1.84_{\pm0.80}}$ | $18.51_{\pm2.99}$ | $\mathbf{18.78_{\pm4.58}}$ |
| | GPan-LoRA | $13.37_{\pm1.38}$ | $5.82_{\pm1.05}$ | $\mathbf{1.33_{\pm0.21}}$ | $\underline{4.68_{\pm1.65}}$ | $\mathbf{2.86_{\pm0.39}}$ | $\mathbf{1.19_{\pm0.83}}$ | $\mathbf{5.03_{\pm1.06}}$ | $\underline{2.74_{\pm0.60}}$ | $\mathbf{12.99_{\pm1.18}}$ | $\underline{18.86_{\pm3.45}}$ |
| NLL (↓) | MLE | $1.56_{\pm0.47}$ | $0.33_{\pm0.01}$ | $0.13_{\pm0.02}$ | $0.63_{\pm0.01}$ | $0.29_{\pm0.01}$ | $0.27_{\pm0.02}$ | $0.77_{\pm0.06}$ | $0.33_{\pm0.01}$ | $1.88_{\pm0.13}$ | $1.75_{\pm0.06}$ |
| | MCD | $1.11_{\pm0.65}$ | $0.46_{\pm0.04}$ | $0.13_{\pm0.03}$ | $0.99_{\pm0.33}$ | $0.49_{\pm0.02}$ | $0.31_{\pm1.04}$ | $0.61_{\pm0.04}$ | $0.24_{\pm0.03}$ | $1.91_{\pm0.09}$ | $1.72_{\pm0.06}$ |
| | DeepEns | $2.03_{\pm0.22}$ | $0.77_{\pm0.20}$ | $0.18_{\pm0.1}$ | $0.94_{\pm0.18}$ | $0.42_{\pm0.01}$ | $0.26_{\pm1.93}$ | $0.38_{\pm0.01}$ | $0.16_{\pm0.01}$ | $1.37_{\pm0.04}$ | $1.26_{\pm0.01}$ |
| | LAP | $\mathbf{0.54_{\pm0.01}}$ | $\mathbf{0.30_{\pm0.04}}$ | $\underline{0.10_{\pm0.01}}$ | $\mathbf{0.44_{\pm0.01}}$ | $\underline{0.24_{\pm0.01}}$ | $0.27_{\pm0.01}$ | $\mathbf{0.28_{\pm0.01}}$ | $\mathbf{0.11_{\pm0.01}}$ | $\underline{1.24_{\pm0.01}}$ | $\underline{1.10_{\pm0.01}}$ |
| | BLoB | $0.79_{\pm0.05}$ | $0.49_{\pm0.19}$ | $0.15_{\pm0.01}$ | $0.78_{\pm0.13}$ | $0.38_{\pm0.01}$ | $0.32_{\pm0.01}$ | $0.55_{\pm0.01}$ | $0.18_{\pm0.02}$ | $2.04_{\pm0.01}$ | $1.87_{\pm0.07}$ |
| | C-LoRA | $\underline{0.68_{\pm0.08}}$ | $\underline{0.40_{\pm0.06}}$ | $0.13_{\pm0.01}$ | $\underline{0.50_{\pm0.02}}$ | $0.29_{\pm0.03}$ | $\mathbf{0.24_{\pm0.01}}$ | $\underline{0.32_{\pm0.04}}$ | $0.13_{\pm0.02}$ | $\mathbf{1.21_{\pm0.15}}$ | $\mathbf{1.06_{\pm0.15}}$ |
| | GPan-LoRA | $0.69_{\pm0.07}$ | $0.47_{\pm0.02}$ | $\mathbf{0.12_{\pm0.01}}$ | $0.46_{\pm0.02}$ | $\mathbf{0.28_{\pm0.01}}$ | $\mathbf{0.24_{\pm0.04}}$ | $0.41_{\pm0.01}$ | $\underline{0.18_{\pm0.01}}$ | $\mathbf{1.20_{\pm0.02}}$ | $\mathbf{1.06_{\pm0.04}}$ |

