# OpenReview forum: "GPan-LoRA: Gaussian Process Amortized Networks for Bayesian Low-Rank Adaptation in Large Language Models"
_ICML.cc/2026/Conference — ICML 2026 regular_

### Official Review · Reviewer_sGsq · 2026-03-04

**Soundness:** 2
**Presentation:** 2
**Significance:** 1
**Originality:** 1
**Overall Recommendation:** 2
**Confidence:** 5

**Summary:**

The paper introduces GPan-LoRA that integrates Gaussian Processes (GPs) with Large Language Models (LLMs); this paper claims that it achieves the balance of scalability and uncertainty estimation

**Compliance With Llm Reviewing Policy:**

Affirmed.

**Final Justification:**

I maintain my final recommendation as **Reject" as I think this paper needs to have a major revision to reach the bar of ICML
For example, the experimental evaluation is limited in **only multiple-choice tasks** and a 8B model
The novelty is very limited.
Therefore, I maintain my final reject recommendation

**Key Questions For Authors:**

1. I cannot discern the balance between computational scalability and uncertainty estimation; in Table 8, GPan-LoRA's inference time of 1.901 seconds is **20.44 times** that of standard deterministic LoRA

2. In this paper, I can tell that the baselines were purposefully set or underestimated for this task

3. Does the posterior need to be resampled for every single generated token? I'm considering the inference latency would multiply exponentially

4. The authors only took experiments on the **8B scale**, which is absolutely not enough; pls at least take some experiments on 30B scale level

**Limitations:**

1. I cannot discern the balance between computational scalability and uncertainty estimation; in Table 8, GPan-LoRA's inference time of 1.901 seconds is **20.44 times** that of standard deterministic LoRA

2. In this paper, I can tell that the baselines were purposefully set or underestimated for this task

3. Does the posterior need to be resampled for every single generated token? I'm considering the inference latency would multiply exponentially

4. The authors only took experiments on the **8B scale**, which is absolutely not enough; pls at least take some experiments on 30B scale level

**Strengths And Weaknesses:**

strength:

1. This paper developed a reliable and scalable uncertainty quantification method in LLMs without full fine-tuning requirement

weakness

 1. I cannot discern the balance between computational scalability and uncertainty estimation; in Table 8, GPan-LoRA's inference time of 1.901 seconds is **20.44 times** that of standard deterministic LoRA

2. In this paper, I can tell that the baselines were purposefully set or underestimated for this task

3. Does the posterior need to be resampled for every single generated token? I'm considering the inference latency would multiply exponentially

4. The authors only took experiments on the **8B scale**, which is absolutely not enough; pls at least take some experiments on 30B scale level

---

> ### Author Rebuttal · Authors · 2026-03-30
>
> We thank the reviewer for the feedback. We believe several concerns stem from misunderstandings about our intended scope of our method and the evaluation setup. We clarify them below point by point.
>
>
> **[Q1] Inference time vs. UQ benefit.** (Response to "Q1")
>
> We would like to clarify that the core contribution of this work focused on introducing **GP–based framework for Bayesian LoRA** that enables **principled UQ in LLMs**, rather than improving the deterministic LoRA. The reported inference overhead corresponds to **uncertainty-aware inference**, which is inherently more expensive than deterministic LoRA due to posterior sampling. Importantly:
>
>
> - The reported cost reflects **multi-sample predictive inference**, which is common in Bayesian methods (including Laplace and ensemble-based approaches). This overhead is **comparable to other UQ baselines** and should not be interpreted as a direct comparison to deterministic inference.
> - The trade-off between computational scalability and uncertainty estimation is common in Bayesian methods. The key scalability advantage of GPan-LoRA is that its Bayesian state is compact and memory-efficient: peak memory is essentially close to MLE (11,363MB vs. 11,324MB), while the Bayesian variables scale as ${O}(lr^2)$ rather than with the full hidden dimension. Moreover, the overhead is not fundamental and can be reduced through implementation improvements (e.g., caching shared activations) and by different model choice (e.g., smaller amortized network).
> - Uncertainty estimation is critical. The additional cost on uncertainty estimates leads to consistently improved calibration (ECE) and NLL in challenging OOD settings (e.g., MMLU-Chem/Phy). For safety-critical UQ, the gains on Chem/Phy are substantial enough to justify this trade-off. The results of selective prediction also show that our method consistently has a lower risk in the high-confidence region when predictive reliability is required. These results are precisely what one expects from a method that trades some aggressiveness in source fitting for significantly more faithful uncertainty estimates.
>
>
> We will clarify in the revision that our use of the word “scalability” mainly refers to Bayesian state size/memory and end-to-end applicability, not only raw latency. We will also clarify this performance-efficiency trade-off and add more rigorous discussion in the revision.
>
>
> **[Q2] Baselines evaluation.** (Response to "Q2")
>
> We respectfully disagree with this concern. We used official implementations whenever available (e.g., LAP and C-LoRA), and otherwise used the same unified Bayesian-PEFT framework, the same base models, the same train/validation split, and the same evaluation protocol. In fact, several baselines outperform GPan-LoRA on raw accuracy in Table 2, suggesting that the baselines are competitive under our setup. We provide detailed implementation descriptions in **Sec. 5.1 and Appendix B.2**, including hyperparameter settings and training procedures for all methods. If the reviewer has **specific concerns regarding particular baselines or configurations**, we would be happy to clarify them and provide more information.
>
>
> **[Q3] Inference latency.** (Response to "Q3")
>
> No. In the multiple-choice QA tasks studied in the paper, one posterior sample corresponds to one stochastic forward pass, and predictive marginalization is linear in the number of samples $S$. There is **no** exponential growth. Moreover, this evaluation protocol is consistent with existing Bayesian LoRA such as **BLoB and C-LoRA**, which adopt the same formulation for classification-style tasks. We will clarify this point in the revision to avoid potential confusion.
>
>
> **[Q4] Backbone.** (Response to "Q4")
>
>
> We thank the reviewer for this concern. Larger-scale evaluation is valuable. However, two distinct 8B backbones already constitute a nontrivial setting for Bayesian fine-tuning, especially because uncertainty evaluation requires repeated posterior sampling. Our experimental setup follows **the most commonly used evaluation regime in prior Bayesian LoRA work**, where baselines are primarily studied at **models up to 7B parameters** (Roberta-base and Llama2-7B in BLoB,  Llama2-7B in C-LoRA). To ensure a fair comparison, we adopt comparable model sizes and further include two recent and advanced 8B models for evaluation.
>
>
> The methodological point of the paper is that the Bayesian component scales in the low-rank adaptation space rather than with the full model width, which is exactly why the approach is promising for larger backbones as well. We will highlight broader-scale evaluation as a next step, but we do not believe the lack of 30B experiments diminishes the novelty or the empirical significance of the current results.

---

> > ### Author Rebuttal · Reviewer_sGsq · 2026-04-04
> >
> > Thank you to the author for the detailed response; this has addressed some of my concerns.
> >
> > However, I still strongly maintain my position to **Reject**
> >
> >
> > 1. Inference latency is often a more critical bottleneck than memory usage during the fine-tuning phase.
> >
> > 2. Regarding the claim that “sampling for classification tasks is linear,” this actually exposes the method’s most critical limitation: evaluation is entirely limited to multiple-choice/true-false questions, completely ignore the true core challenge of LLMs and real-world scenarios, open-ended text generation.
> >
> > 3. I suspect this method simply draws inspiration from similar approaches and datasets presented in other papers, then borrows some concepts from Bayesian theory to come up with a new method.

---

> > > ### Author Response · Authors · 2026-04-05
> > >
> > > We thank the reviewer for the follow-up comments. We believe several concerns stem from misunderstandings about the intended scope of our method and the evaluation perspective. We clarify them below.
> > >
> > >
> > > **1. On latency vs. memory.**
> > >
> > > We emphasize that **our comparison is within the context of uncertainty quantification methods, particularly Bayesian approaches**, where additional computational overhead is inherent to modeling predictive uncertainty (e.g., posterior sampling). Therefore, the relevant baseline is not deterministic fine-tuning, but prior UQ methods such as LAP, BLoB, and C-LoRA.
> > >
> > >  Within this context, **our method achieves substantially improved uncertainty quality while maintaining comparable inference latency.** This demonstrates a more favorable trade-off between computational cost and UQ performance.
> > >
> > > In addition, our lightweight variant (LL GPan-LoRA) achieves lower inference latency than other variational inference-based baselines (e.g., BLoB and C-LoRA), while preserving strong uncertainty estimation performance.
> > >
> > >
> > > **2. On MCQA evaluation.**
> > >
> > > We believe this concern arises from a difference in evaluation perspective. **Multiple-Choice Question Answering (MCQA) is a standard and controlled setting for uncertainty quantification, widely adopted in prior and recent Bayesian LoRA work** (e.g., LAP, BLoB, C-LoRA), as it enables well-defined evaluation of calibration metrics such as ECE, NLL, and selective prediction.
> > >
> > > Uncertainty estimation for open-ended text generation is indeed an important challenge for LLMs, but it is beyond the scope of this work.
> > > Importantly, **uncertainty quantification for common-sense reasoning under the MCQA setting is non-trivial.** Prior studies have shown that even in structured prediction settings such as MCQA, LLMs often exhibit significant overconfidence, including in safety-critical applications such as healthcare [1].
> > > This further highlights the importance and practical relevance of our work.
> > >
> > >
> > > **3. On novelty.**
> > >
> > > We note that this concern is not supported by specific references or identified technical overlap. Our work is to show how GP function-space modeling can be recast through LoRA’s low-rank map and introduce a scalable Bayesian adapter whose uncertainty lives in the low-rank latent space rather than the full feature space. This is what makes end-to-end GP-based Bayesian LoRA possible for LLM fine-tuning while preserving the parameter efficiency. Along this way, GPan-LoRA introduces a GP-induced functional-space prior/posterior, together with amortized variational inference and shared mean function in the low-rank adaptation space.
> > >
> > > **The novelty of this paper lies in the new Bayesian LoRA framework in GP functional space, which differs fundamentally from prior Bayesian LoRA approaches.**
> > >
> > > [1] Kim et al., Limitations of Large Language Models in Clinical Problem-Solving Arising from Inflexible Reasoning. Nature, 2025.

---

### Official Review · Reviewer_rho8 · 2026-03-10

**Soundness:** 3
**Presentation:** 3
**Significance:** 3
**Originality:** 3
**Overall Recommendation:** 5
**Confidence:** 3

**Summary:**

This paper addresses an unresolved challenge: the inherent trade-off between scalability and uncertainty quantification (UQ) quality in existing Bayesian LoRA methods for large language models (LLMs). Scalable approaches in this space often suffer from poorly calibrated uncertainty and severe overconfidence under distribution shift, while theoretically principled UQ methods incur prohibitive computational costs when applied to large-scale LLMs. To address this issue, the authors propose GPan-LoRA, a scalable Gaussian Process (GP)-based Bayesian LoRA framework that combines sparse GP approximation with amortized variational inference. By confining Bayesian inference to the low-rank latent space of LoRA and establishing the equivalence between sparse GP and variational Bayesian network layers, the method preserves the principled Bayesian function prior of GPs while maintaining computational efficiency. Altogether, this work investigates a high-impact dimension of principled UQ for parameter-efficient LLM fine-tuning, with rigorous theoretical support for its core design and extensive experiments verifying its advantages in uncertainty calibration, especially under distribution shift.

**Compliance With Llm Reviewing Policy:**

Affirmed.

**Ethical Review Flag:**

Flag this paper for an ethics review.

**Final Justification:**

This paper presents GPan-LoRA, an expandable Bayesian LoRA framework based on Gaussian processes. It achieves scalable and principled uncertainty quantification through sparse GP approximation and amortized variational inference. This work has a solid theoretical foundation (Theorem 4.1 rigorously proves the exact reconstruction of GP through LoRA linear transformation), and it is the first scalable Bayesian LoRA framework based on Gaussian processes, fundamentally shifting the modeling approach from prior distributions on weight parameters to prior distributions on function spaces.
The author's rebuttal comprehensively and constructively addressed all my initial concerns, including the selection of the kernel, the multi-layer GP design, the details of the amortized network, the inductive bias of weight sharing, the diversity of tasks, and the distinction of novelty. The rebuttal significantly enhanced my confidence in the completeness of the work, changed my initial assessment, and convinced me that the paper met the acceptance criteria of ICML. In conclusion, I strongly recommend the acceptance of this paper.

**Key Questions For Authors:**

1. The paper mainly adopts the stationary SE kernel in implementation, could you explain the reason for this kernel selection, and discuss the impact of different types of kernel functions on the performance and computational complexity of GPan-LoRA?
2. When applying GPan-LoRA to multiple LoRA layers of Transformer, what assumptions does the method make for the multi-layer GP modules, and how does it avoid the high complexity of deep Gaussian processes in uncertainty propagation?
3. The amortized network is a core component of the method, but its specific structural design is rarely mentioned in the main text. Could you elaborate on the design details of the amortization network and the weight sharing mechanism with the LLM feature extractor?
4. The ablation study verifies that the weight sharing strategy is critical to the model performance, could you explain the inductive bias brought by this design and whether it will limit the flexibility of the model in mean fitting and uncertainty modeling?

**Limitations:**

The authors only briefly mention the positive social impact of the work in the Impact Statement, but do not systematically discuss the inherent limitations of the method design, such as the constraints of kernel selection and multi-layer GP assumptions. In addition, the paper does not analyze the potential negative social impact of the work, such as the decision-making risks that may still exist in high-stakes scenarios even with improved uncertainty calibration. It is suggested that the authors supplement the discussion of the method's limitations and relevant ethical considerations in the revision.

**Strengths And Weaknesses:**

### Soundness
The core technical design of the method is supported by solid theoretical foundations, with a rigorous proof of Theorem 4.1 that guarantees the exact GP recasting through LoRA's linear transformation, laying a solid theoretical basis for constructing GP in the low-rank space. The authors also derive the mathematical equivalence between sparse GP approximation and variational Bayesian network layers, which effectively solves the incompatibility between traditional GP inference and LLM training pipelines, and the key technical claims are consistently supported by ablation studies and comparative experiments. However, the method design lacks sufficient analysis of kernel function selection, with no discussion on the applicability of different types of kernels and their impact on model performance; the processing of uncertainty propagation in multi-layer GP modules is not clearly explained, and the rationality of the mean-field Gaussian posterior assumption and the design of inducing points are not elaborated, which affects the completeness of the technical elaboration.

### Presentation
The paper has a clear overall structure and smooth narrative logic, with a well-organized layout from research background, methodology to experimental verification, which conforms to the writing norms of top machine learning conferences. The module structure comparison diagram intuitively shows the core innovation of the method, and the related work section accurately sorts out the research progress and clearly locates the innovation of this work. Nevertheless, the main text lacks detailed description of the key design details of core components such as the amortized network, and some content presentation is imperfect, such as the incomplete filling of Table 1 and inconsistent symbol definitions in different parts, which brings inconvenience to reading and understanding.

### Significance
This work addresses an important demand for trustworthy AI deployment, as principled and efficient UQ is crucial for the application of LLMs in high-stakes scenarios such as healthcare and finance. The proposed framework opens up a new direction of introducing GP function space prior into Bayesian LoRA, which has important reference value for subsequent research on Bayesian parameter-efficient fine-tuning. Meanwhile, the method maintains lightweight computational overhead while achieving excellent UQ performance, and provides a lightweight last-layer variant, which has good practical application potential. However, the current verification is only limited to 8B scale models and multi-choice QA tasks, with no discussion on the scalability to larger models and the adaptability to mainstream LLM tasks such as text generation, which limits the demonstration of the method's wide applicability.

### Originality
The work has clear core originality, as it is the first scalable GP-based Bayesian LoRA framework, which fundamentally changes the modeling idea of existing Bayesian LoRA methods from weight parameter prior to function space prior, and is essentially different from previous related works. The combination of sparse GP, amortized variational inference and LoRA mechanism is innovative, and the design directly targets the overconfidence defect of existing Bayesian LoRA methods, rather than a simple combination of existing technologies. That said, the paper does not clearly distinguish the incremental innovation of some components from existing work such as amortized variational deep kernel learning, nor does it deeply analyze the essential difference between the proposed amortized design and the contextual module of C-LoRA, which weakens the elaboration of the method's novelty.

---

> ### Author Rebuttal · Authors · 2026-03-30
>
> We sincerely thank the reviewer for the constructive and thoughtful feedback. We appreciate your recognition of the theoretical soundness, clear presentation, and originality of our approach, as well as your insightful questions on kernel design, multi-layer modeling, and amortized inference.
>
>
> **[Q1] Kernel choice selection** (response to "different types of kernels" and "Q1")
>
> Our method is kernel-agnostic; SE is used as a robust default. The SE kernel is the most commonly used in GP-based models and works reliably in high-dimensional feature spaces induced by LLM representations. Other kernels (e.g., Matérn) can be incorporated easily by replacing SE kernel with any tractable PSD kernel on the projected space.
>
> Due to space constraints, we kindly refer the reviewer to our response to **Reviewer Q9Pq (Q2)** for further details on kernel choices and sensitivity.
>
>
> **[Q2] Multi-layer GP modules and avoiding deep GP complexity.** (Response to "uncertainty propagation in multi-layer GP modules" and "Q2")
>
> Each LoRA-equipped Transformer layer has its own local GPan adapter acting on that layer's projected feature. We do **NOT** perform exact deep-GP inference over all layer outputs. Instead, each local GP module is approximated by a finite-dimensional Bayesian adapter via the sparse GP/VBN formulation (Eq 14-16), and uncertainty is propagated by standard stochastic forward passes through sampled low-rank $Z$. In practice, this corresponds to a variational posterior over the per-layer Bayesian coefficients, not a full exact DGP posterior over all hidden states. This is exactly how we avoid the intractable complexity of deep GPs while preserving the GP functional prior in each low-rank feature space and flexible BNN training techniques.
>
>
> **[Q3] Amortization network details.** (Response to  "design details of amortized network" and "Q3")
>
> The amortization is intentionally lightweight: it takes the projected feature $Ax$ as input and outputs the variational posterior parameters of the low-rank Bayesian coefficients (the mean and log-variance of $q_\phi(Z|x)$). It shares the same feature extractor and a lightweight feed-forward (2 layer MLP), so we do not introduce a separate heavy contextual model over the full hidden state. The role of the amortized network is to further alleviate the overconfidence, which is discussed in Appendix 5. The core contribution of this work still focused on introducing the GP–based framework for Bayesian LoRA that enables principled UQ in LLMs. We will add these details, along with pseudocode, in the revision to make the pipeline easier to follow.
>
>
> **[Q4] Weight sharing inductive bias.** (Response to "Q4")
>
> The weight sharing ties the deterministic mean-fitting term and the stochastic uncertainty term to the same task-relevant low-rank subspace. This imposes a useful inductive bias: directions that matter for adaptation are also the directions along which uncertainty is modeled, which improves statistical efficiency and prevents the posterior from drifting into mislead directions in small-data PEFT. Importantly, this does *not* eliminate flexibility, because the Bayesian adapter still acts on nonlinear kernel features $\kappa(Ax)$ through the random matrix $Z$. The ablation in Table 3 confirms that this bias is beneficial rather than restrictive: removing weight sharing degrades both accuracy and calibration, especially under large shift.
>
>
> **[5] Response to other concerns not mentioned in above questions**
>
> - Mean-field Gaussian posterior assumption: The mean-field variational posterior aligns the GP induced prior. The kernel activation creates an orthonormal decomposition of GP where the weight $z_i$ in Eq (16) is i.i.d. Therefore, it is reasonable assuminge the independent but not identical variational weights.
>
> - Design of inducing points: Our variational GP is not a looser GP approximation than inducing SKI-GP methods[1]. Therefore, the design of inducing points does not cause any degradation from existing inducing-point methods. Due to the page limitation, we did not include this detailed design in the main body, but we will elaborate the inducing point design in the revised version.
>
> - Scalability to large backbone: Larger-scale evaluation is valuable. However, two distinct 8B backbones already constitute a nontrivial setting for Bayesian fine-tuning, especially because uncertainty evaluation requires repeated posterior sampling. Our experimental setup follows **the most commonly used evaluation regime in prior Bayesian LoRA work**, where baselines are primarily studied at **models up to 7B parameters** (Roberta-base and Llama2-7B in BLoB,  Llama2-7B in C-LoRA). To ensure a fair comparison, we adopt comparable model sizes and further include two recent and advanced 8B models for evaluation.
>
> [1] Wilson et al., Kernel interpolation for scalable structured Gaussian processes.ICML, 2015.

---

> > ### Author Rebuttal · Reviewer_rho8 · 2026-04-06
> >
> > The authors' rebuttal has adequately addressed several of my core concerns, including kernel choice, multi-layer GP complexity, weight sharing inductive bias, and mean-field posterior justification.  However, there remain partially unresolved issues that prevent a full upgrade of my score:
> >
> > 1. **Amortized network details**: The current description of this core component is still insufficient for full reproducibility, even with the promise of future revisions.
> > 2. **Scalability & task diversity**: The evaluation remains limited to multiple-choice QA, with no evidence on generative tasks where uncertainty quantification is most critical.
> > 3. **Novelty differentiation**: The rebuttal fails to clearly distinguish the proposed amortized design from closely related prior work, weakening the novelty narrative.

---

> > > ### Author Response · Authors · 2026-04-07
> > >
> > > We sincerely thank the reviewer for the positive assessment and for acknowledging that our rebuttal has addressed several core concerns. Below, we provide additional clarifications on the remaining points to further improve clarity and completeness.
> > >
> > >
> > > **1. Amortized network details.**
> > >
> > >
> > > We thank the reviewer for this constructive suggestion. As mentioned in our rebuttal, the amortized network is implemented as a lightweight **two-layer MLP that maps LoRA features of dimension $r$ to variational parameters**. Specifically, it consists of a linear layer, followed by a GELU activation, and a second linear layer that outputs a vector of size $r^2 \times l$, corresponding to the parameterization of the induced low-rank GP structure, where $l$ is the size of inducing points.
> > >
> > > We use two such networks to amortize the mean and covariance. The detailed implementation is **open-sourced in the attached Supplementary Material (*gpan_lora.py*)**.   We will clarify the design of the amortized network in the revision and include pseudocode in the appendix to further improve clarity and reproducibility.
> > >
> > > **2. Scalability & task diversity.**
> > >
> > >
> > > We thank the reviewer for this important point. We agree that extending uncertainty quantification to open-ended text generation is a valuable direction. In this work, **we adopt the MCQA setting as a controlled and widely used evaluation protocol in prior Bayesian LoRA literature** (e.g., LAP, BLoB, C-LoRA), as it enables well-defined and rigorous evaluation of uncertainty metrics (e.g., ECE and NLL), and selective prediction.
> > >
> > >
> > > Importantly, **uncertainty quantification for common-sense reasoning under the MCQA setting is non-trivial.** Prior studies have shown that even in structured prediction settings such as MCQA, LLMs often exhibit significant overconfidence, including in safety-critical applications such as healthcare [1]. This further highlights the importance and practical relevance of our work.
> > >
> > >
> > > Moreover, **our method is not inherently limited to MCQA.** Since it performs posterior sampling over model parameters, it can be extended to open-ended generation by sampling multiple outputs and constructing an empirical predictive distribution over extracted answers, provided that an appropriate answer extraction or normalization scheme is defined.
> > >
> > >
> > > We view large-scale evaluation on generative tasks as an important but orthogonal direction to our primary focus on developing an effective Bayesian LoRA framework for uncertainty quantification. This does not limit the applicability of our method, and we will clarify this extension in the revision.
> > >
> > > **3. Novelty differentiation.**
> > >
> > > The roles of the amortized network in GPan-LoRA and the contextual module in C-LoRA are conceptually different. Specifically, **amortization in GPan-LoRA is to tie the variational parameters with deterministic parameters, which introduces an implicit regularization effect on the latter as well**. In contrast, C-LoRA uses a contextual module to capture data-dependent uncertainty. While C-LoRA shares the LoRA matrices $A$ and $B$ to maintain parameter efficiency, resulting in a similar engineering architecture, the underlying modeling principles differ.
> > >
> > >
> > > We would like to emphasize that **the top priority of this paper lies in the new Bayesian LoRA framework in GP-induced functional space**. Our work is to show how GP function-space modeling can be recast through LoRA’s low-rank map and introduce a scalable Bayesian adapter whose uncertainty lives in the low-rank latent space rather than the full feature space. **Directly combining GP with deep neural networks [2] or using a contextual network without GP could still suffer from the overconfidence issue (e.g., C-LoRA as illustrated in Fig. 1)**, since expressive likelihood strongly favors sharp posterior modes that fit the data closely. Along this way, the amortized network serves as an auxiliary regularizer to alleviate the overconfidence caused by expressive likelihood.
> > >
> > >
> > > We thank the reviewer again for the valuable feedback. We will incorporate these suggestions to further improve the clarity and presentation of the paper in the revision.
> > >
> > >
> > >
> > > [1] Kim et al., Limitations of Large Language Models in Clinical Problem-Solving Arising from Inflexible Reasoning. Nature, 2025.
> > >
> > >
> > > [2]  Ober et al., The promises and pitfalls of deep kernel learning. UAI, 2021.

---

### Official Review · Reviewer_ASEr · 2026-03-11

**Soundness:** 3
**Presentation:** 3
**Significance:** 2
**Originality:** 3
**Overall Recommendation:** 4
**Confidence:** 4

**Summary:**

The paper introduces an uncertainty estimation method for parameter efficient fine-tuning based and amortized GP that is injected in between LoRA matrices. The method performs competitively against various baselines from the literature on a range of language classification benchmarks with Llama3.1 and Qwen3 8B architectures per the usual uncertainty estimation metrics on in- and out-of-distribution tasks.

**Compliance With Llm Reviewing Policy:**

Affirmed.

**Final Justification:**

Overall, the paper makes a strong enough case to me with its theoretical contributions and empirical results for acceptance, but not decisively enough to warrant a higher score.

**Key Questions For Authors:**

See first point of weaknesses in particular.

**Limitations:**

yes

**Strengths And Weaknesses:**

Strengths:
* The method is novel as far as I am aware and theoretically well-grounded
* The experiments are thorough (range of tasks and settings, two architectures, ablations)
* Calibration appears to be good quite consistently
* The method also performs well in the selective prediction setting, although it would be helpful to have AUC scores for the in-distribution risk in particular (Fig 3)

Weaknesses:
* I'm a bit confused by the discussion around scalability and Table 8. How is there a 30x computational overhead for a size 3 ensemble? Surely for LoRA parameters these don't need to be offloaded to CPU? Also I'm a bit surprised that Laplace is marked as not scalable in Table 1 when it is twice as fast as the proposed method. I understand that memory is an issue, but to a degree that is an implementation choice (you can always sample multiple parameters, but then you're back to having an ensemble of course)
* Clarity could be improved by adding pseudo-code for training and inference
* Performance is overall a bit of a mixed bag, but that tends to be the case with uncertainty estimation. There seems to be a loss of raw accuracy overall, however.
* Fig 1 is missing a HMC baseline/ground truth

---

> ### Author Rebuttal · Authors · 2026-03-30
>
> We sincerely thank the reviewer for the constructive and thoughtful feedback. We appreciate your recognition of the novelty and soundness of our approach, the comprehensive experimental evaluation, and the consistently strong calibration performance across settings.
>
>
> **[W1] Scalability discussion** (Response to "W1")
>
>
> We thank the reviewer for pointing this out.
>
>
> - Scalability in Bayesian LoRA. The inference time in Table 8 reports end-to-end latency cost under the full evaluation step and the same test set, which can be caused by multiple engineering factors. Our qualitative “scalability'” score in Table 1 refers to compact Bayesian state and memory scaling rather than raw latency alone. GPan-LoRA achieves peak memory essentially close to MLE (11,363MB vs. 11,324MB), while the Bayesian variables scale as ${O}(lr^2)$ rather than with the full hidden dimension. In practice, the end-to-end latency can be optimized and reduced through implementation improvements (e.g., caching shared activations, optimized buffer usage) and by different model choice (e.g., smaller amortized network), which provides a more flexible performance trade-off.
> - Deep Ensemble and Laplace-LoRA. For DeepEns, the reported end-to-end latency reflects the wall-clock time of the full evaluation step in our implementation, rather than the pure forward latency of three ensemble members alone. In particular, the timing includes sequential evaluation of the ensemble members by switching LoRA adapters, as well as aggregation overhead. For LAP, qualitative “scalability” score in Table 1 refers to memory and end-to-end applicability rather than raw latency alone. Even in the current 8B setting, Laplace-LoRA requires 33,121MB peak memory (2.93×MLE) due to second-order state, whereas GPan-LoRA stays essentially at MLE memory. Importantly, this result corresponds to the **recommended and default configuration** of Laplace-LoRA, where a Kronecker-factored (KFAC) approximation is used, as in the original implementation. While lighter approximations (e.g., diagonal Hessian) could reduce memory, they come with a clear trade-off: a significantly weaker posterior approximation, which can degrade uncertainty estimation quality. We will revise the text/caption to distinguish the scalability of DeepEns and LAP more clearly.
> - The lightweight *LL GPan-LoRA* variant already shows the intended practical trade-off: it reduces latency from 1.901s to 0.935s (0.78x BLoB, 0.77x C-LoRA), while still remaining much better calibrated UQ than BLoB/C-LoRA on the hardest shifts (Chem/Phy ECE =14.37/13.67 (GPan) vs. 24.45/32.16 (BLoB) and 19.39/25.68 (C-LoRA).
>
>
> **[W2] Clarity suggestion** (Response to "W2")
>
>
> We thank the reviewer for this helpful suggestion. We will provide pseudo-code for both the training and inference procedures in the final version to improve clarity and reproducibility.
>
>
> **[W3] Performance discussion** (Response to "W3")
>
>
> We thank the reviewer for this thoughtful insight. We would like to emphasize that this is standard in uncertainty-estimation papers: the central question is whether the method produces **reliable predictive probabilities**, especially under shift. GPan-LoRA is strongest exactly in that regime. Table 2 presents the results for in-distribution (ID) and out-of-distribution (OOD) settings. In the ID setting, the accuracy of our proposed method is highly competitive relative to the baseline (i.e., the best on ARC-E and very close to the best on almost all other datasets). For the large OOD shifts, it consistently improves ECE and NLL by a large margin; we only find some ACC drops on 1 datasets under the small distribution shift. The results in Sec.5.3 show that our method consistently has a lower risk in the high-confidence region when predictive reliability is required. These results are precisely what one expects from a method that trades some aggressiveness in source fitting for significantly more faithful uncertainty estimates.
>
>
> **[W4] Presentation suggestion** (Response to "W4")
>
>
> We thank the reviewer for this suggestion. Fig. 1 is intended as a **compact qualitative illustration comparing practical Bayesian LoRA methods**. The more relevant oracle in this toy setting is the exact GP posterior (ground truth) under the data-generating kernel, which we already **include in Appendix B.1/Fig. 5**. We will include the ground truth more explicitly in the main-text discussion and figures.

---

> > ### Author Rebuttal · Reviewer_ASEr · 2026-04-04
> >
> > Thank you for responding in detail to all points. I will base an adjustment of the score on the reviewer discussion.

---

### Official Review · Reviewer_Q9Pq · 2026-03-13

**Soundness:** 3
**Presentation:** 3
**Significance:** 2
**Originality:** 3
**Overall Recommendation:** 4
**Confidence:** 4

**Summary:**

GPan-LoRA proposes integrating GP-based uncertainty quantification into LoRA fine-tuning via sparse variational GPs combined with amortized variational inference. The key insight is that GP inference can be exactly recast through LoRA's low-rank linear projection (Theorem 4.1). An amortized network produces input-dependent variational parameters, mitigating the overconfidence commonly observed in standard variational approaches. Experiments on LLaMA3.1-8B and Qwen3-8B across eight datasets show strong calibration, particularly under large distribution shift.

**Compliance With Llm Reviewing Policy:**

Affirmed.

**Key Questions For Authors:**

1. Theorem 4.1 establishes exact GP recasting under linear transformations, but the subsequent VBN approximation in Section 4.2 introduces additional approximation error. Can the authors quantify or bound the gap between the exact GP posterior and the final approximation used in practice?
2. The method focuses exclusively on SE kernels. How sensitive are the results to kernel choice, and can the framework accommodate non-stationary kernels that may be more appropriate for language data?

**Limitations:**

yes

**Strengths And Weaknesses:**

Strengths:
1. Theorem 4.1 formally establishes that GP inference can be recast through LoRA's linear projection, which is a non-trivial theoretical contribution.
2. Under large distribution shifts (MMLU-Chem/Phy), GPan-LoRA achieves dramatically better ECE than all baselines.

Weaknesses:
1. In Table 2, GPan-LoRA frequently underperforms on accuracy, sometimes substantially. The paper frames this as a calibration-accuracy tradeoff, but the gap is large enough to raise concerns about whether the Bayesian regularization is too aggressive.
2. Table 8 shows GPan-LoRA has 20.44× inference time overhead over MLE, significantly worse than BLoB and C-LoRA. For practical deployment this is a serious limitation. The lightweight LL GPan-LoRA variant (10.05×) partially addresses this but at meaningful performance cost.

---

> ### Author Rebuttal · Authors · 2026-03-30
>
> We appreciate the reviewer’s constructive feedback and the recognition of our theoretical contribution and improved calibration under distribution shifts. Below, we respond to the key suggestions and concerns.
>
>
> **[W1] Calibration accuracy trade-off**
>
> Table 2 presents the results for in-distribution and out-of-distribution settings. In the in-distribution setting, the accuracy of our proposed method is highly competitive relative to the baseline (i.e., the best on ARC-E and very close to the best on almost all other datasets). For the out-of-distribution setting, we only find some drops under the small distribution shift. Besides, the results in Sec. 5.3 show that our method consistently has a lower risk when extremely reliable predictions are required.
>
>
> **[W2] Inference time overhead**
>
> - **Inference time in Table 2 is the end-to-end latency cost** under the full evaluation step and the same test set, which can be **caused by multiple engineering factors**. The key scalability advantage of GPan-LoRA is that its Bayesian state is compact and memory-efficient: peak memory is essentially close to MLE (11,363MB vs. 11,324MB), while the Bayesian variables scale as ${O}(lr^2)$ rather than with the full hidden dimension. We agree that the end-to-end latency is also important to optimize in practice and can be reduced through implementation improvements (e.g., caching shared activations, optimized buffer usage) and by different model choice (e.g., smaller amortized network), which provides a more flexible efficiency–performance trade-off.
> - The lightweight *LL GPan-LoRA* variant already shows the intended practical trade-off: it reduces latency from 1.901s to 0.935s (0.78x BLoB, 0.77x C-LoRA), while still remaining much better calibrated UQ than BLoB/C-LoRA on the hardest shifts (Chem/Phy ECE =14.37/13.67 (GPan) vs. 24.45/32.16 (BLoB) and 19.39/25.68 (C-LoRA).
>
>
> **[Q1] Approximation gap discussion**
>
> Theorem 4.1 itself introduces no approximation error: the recasting through the LoRA map is exact. The practical approximation appears only in Sec. 4.2, where we use an inducing sparse variational GP and then amortize the variational posterior. The approximation gap of GPan consists of two parts:
>
>
> - Inducing-points approximation. Let $V_U = \mathrm{span}( k(\cdot,u): u\in U )$. Our decomposed $\kappa(Ax)$ is an orthonormal basis of $V_U$, so $k(Ax,U)K(U,U)^{-1}g(U)=\sum_{j=1}^r \kappa_j(Ax) z_j$ is the same inducing-point GP approximation. It can be shown that the best approximation gap that we can achieve is ${O}(l^{-1})$ [1], where $l$ is the number of inducing points.
> - Amortized variational inference, which is a common optimization gap in variational Bayesian inference.
>
> [1] Bach, F. On the equivalence between kernel quadrature rules and random feature expansions. JMLR, 2017.
>
> **[Q2] Kernel choice selection and sensitivity**
>
> - Generalization to other kernels. First, we would like to emphasize that our primary contribution is not tied to a specific kernel choice, but rather lies in introducing a scalable GP-based Bayesian formulation into LoRA via low-rank projection and amortized inference. Theorem 4.1 is kernel-agnostic: **any kernel that admits a representation in the projected space can be incorporated into our formulation**. We use the SE kernel because it is the most stable/default choice for sparse GP training and because, after the learned low-rank projection $Ax$, an isotropic SE in latent space already corresponds to a nontrivial anisotropic kernel in the original representation space.
> - Sensitivity to kernel choices. In general, the kernel choice can depend on the specific task we need to solve. However, in LLM fine-tuning, the kernel is applied to learned LLM features rather than raw text, so the choice is less restrictive than it may appear. Regarding non-stationary kernels, our framework is flexible and can accommodate more expressive kernel functions, including non-stationary ones, as long as they can be defined over the projected feature space in Eq. (14-16). We agree this is a promising direction, particularly for language data, and will include a discussion of this extension in the revised version.
> - Additional ablations regarding kernel choices. We conducted additional experiments replacing the SE kernel with a Matérn kernel. The results below show that performance differences are minor, indicating that our method is relatively robust to the choice of kernel.
>     | |WG-S|ARC-C|ARC-E|WG-M|OBQA|BoolQ|ARC-C(OOD)|ARC-E(OOD)|Chem(OOD)|Phy(OOD)|
>     |-|-|-|-|-|-|-|-|-|-|-|
>     |ACC|74.13(2.61)|81.19(1.86)|91.96(0.66)|78.08(0.70)|85.53(0.75)|88.63(0.49)|76.23(2.44)|83.21(0.56)|48.66(4.50)|39.33(2.08)|
>     |ECE|12.82(3.20)|5.48(1.70)|2.76(0.82)|5.00(0.61)|2.77(0.17)|1.38(0.38)|6.18(2.19)|3.62(0.43)|12.58(2.40)|17.76(2.49)|
>     |NLL|0.72(0.11)|0.49(0.02)|0.23(0.01)|0.48(0.01)|0.38(0.04)|0.27(0.01)|0.58(0.01)|0.45(0.02)|1.19(0.05)|1.25(0.01)|

---

> > ### Author Rebuttal · Reviewer_Q9Pq · 2026-04-03
> >
> > Thanks for your reply, and all my concerns have been addressed.

---

### Decision · Program_Chairs · 2026-04-30

**Decision:**

Accept (regular)

**Comment:**

GPan‑LoRA seems a strong and quite original paper that is quite close to the bar of a solid accept. With a bit more statistical rigour around the empirical evidence (which I emphasise as decisive for all borderline papers I overlook), the paper is fully publishable.

The paper tackles an important problem: scalable and calibrated inference for Bayesian LoRA for LLM. It enables a GP view on the low-rank adaptations directly. Essentially, the authors define a GP functional prior in the LoRA low-rank space, then approximate the posterior with a sparse GP approximation. This seems like an elegant solution to an important problem of UQ from LLMs.

Reviewers are mainly concerned with the accuracy-calibration trade-offs, as it seems (although, as mentioned above, not statistically sound) that GPan‑LoRA sometimes loses a bit of accuracy, yet with somewhat better calibration (again, not statistically sound). Also, the reviewers question the meaning of “scalability” given higher latency than some baselines, and I somewhat agree with what is shown in Table 8 it seems that neither inference time nor memory is improved compared to baselines (again, hard to claim significance here), although I would not find this issue critical for acceptance. Some reviewers complain about the lack of sensitivity to the kernel choice, multi‑layer GP design, etc., so clear ablations on these parts would be appreciated.  Three reviewers consider their concerns fully or largely resolved and stay at weak accept/accept; one remains somewhat skeptical, despite the authors giving detailed and sufficiently good (in my opinion) replies to their critical comments.

To sum up, as the only sound weakness, I would emphasises there are some limitations with experimental design and statistical evaluation (although that is the case for the majority of the ICML submissions). On the other hand, the paper is original, relevant, technically sound, and well written. The topic is timely and highly relevant for ICML community. Rebuttal is also strong and detailed.  Based on the tradeoff between all reviews (accept, weak accept, weak accept and reject): a weak accept is recommended.

If the paper is accepted, after all, I expect to see statistical evaluations relevant to experimental design: pairwise tests, or appropriate mixed-effect regression models should be chosen to claim significance of the results, as the current evaluation with only general summaries (Tables 1 - 9) is not rigorous in that sense, and thus are hard to draw sound empirical evidence on. I believe it will strengthen the claims significantly and ensure a fully sound contribution is published. Thus, If the paper is accepted, I expect the proper statistical evaluation in the final version.